# Global hotspots for the occurrence of compound events

Nina N. Ridder [1✉], Andy J. Pitman [1], Seth Westra[2], Anna Ukkola [3], X. Do Hong [4,5], Margot Bador [1], Annette L. Hirsch [1], Jason P. Evans [1], Alejandro Di Luca [1] & Jakob Zscheischler [6,7]

Compound events (CEs) are weather and climate events that result from multiple hazards or drivers with the potential to cause severe socio-economic impacts. Compared with isolated hazards, the multiple hazards/drivers associated with CEs can lead to higher economic losses and death tolls. Here, we provide the first analysis of multiple multivariate CEs potentially causing high-impact floods, droughts, and fires. Using observations and reanalysis data during 1980–2014, we analyse 27 hazard pairs and provide the first spatial estimates of their occurrences on the global scale. We identify hotspots of multivariate CEs including many socio-economically important regions such as North America, Russia and western Europe. We analyse the relative importance of different multivariate CEs in six continental regions to highlight CEs posing the highest risk. Our results provide initial guidance to assess the regional risk of CE events and an observationally-based dataset to aid evaluation of climate models for simulating multivariate CEs.

[1] Australian Research Council Centre of Excellence for Climate Extremes, University of New South Wales, Sydney, NSW, Australia. [2] School of Civil, Environmental and Mining Engineering, University of Adelaide, Adelaide, SA, Australia. [3] Australian Research Council Centre of Excellence for Climate Extremes, Australian National University, Canberra, ACT, Australia. [4] School for Environment and Sustainability, University of Michigan, Ann Arbor, MI, USA. [5] Faculty of Environment and Natural Resources, Nong Lam University, Ho Chi Minh City, Vietnam. [6] Oeschger Centre for Climate Change Research, University of Bern, Bern, Switzerland. [7] Climate and Environmental Physics, University of Bern, Bern, Switzerland. ✉email: n.ridder@unsw.edu.au

Extreme weather and climate events often result from a combination of multiple hazards or drivers (a driver is a direct cause of climate-related hazards, see Table 1 in ref. [1]). These events are often referred to as compound events (CEs)[1–4]. The interaction of multiple hazards and/or drivers that generates CEs often lead to more severe ecologically and socioeconomically damaging events compared to single hazard events[5]. One specific class of CEs, multivariate CEs[3], occurs when two or more drivers and/or hazards impact a region simultaneously. This joint occurrence will often exacerbate the impacts compared to individual hazards occurring in isolation. One example of a high-impact multivariate CE was the 2012 Groningen event, where extreme inland water levels were caused by elevated coastal water levels, preventing runoff of high rainfall for several tidal periods[6]. A more recent example was the increase in fire danger in eastern Australia during spring and summer (Sep–Jan) 2019–2020 due to the simultaneous occurrence of high temperatures, drought conditions, high fuel load, and strong winds. Here, we use the risk framework of the Intergovernmental Panel for Climate Change (IPCC), and define climate extremes as the occurrence of a value of a weather/climate variable within either tail of the variable's observed distribution[4]. This means that not all occurrences of a CE necessarily lead to an impact as these are dependent on a combination of hazard occurrence, as well as vulnerability and exposure of the affected region/system. Our analysis focusses on the occurrence of CEs defined as the joint probability of two hazards, and we do not explore whether the CEs necessarily lead to impacts. For multivariate risk assessments that require CEs to have impacts, our results can be interpreted as the climatology of the precursors to CEs, or of potential CEs. CEs are by definition events with multiple, potentially interacting, meteorological processes, and consequently require different analysis methods compared with their univariate counterparts. For instance, the probability of the near flood event in Groningen in 2012 or compound hot and dry summers is widely underestimated, when classical univariate statistical methods are used[6,7].

While analyses of univariate climate extreme events is common (e.g., extreme rainfall[8,9], heatwaves (HWs)[10,11], extreme temperatures[8], and flood[12]) there has been little research on the probability of multivariate CEs at larger scales, with the exception of hot and dry events and compound flooding[6,13–18]. Previous studies have analysed specific hazard pairs and used climate models to account for sparse data. There are uncertainties around whether coarse resolution models can reproduce the physical relationships associated with multivariate CEs, which raises doubts over the reliability of these single hazard pair studies[6,7,17,19]. No previous analysis has examined correlations between a range of hazards, or the geographic regions where different multivariate CEs are most likely to occur. Instead, studies have focused on describing specific events, the influence of correlation on return periods[7], and, with the exception of a few isolated studies[20], regional scales. This lack of a global geographical climatological fingerprint of multivariate CEs may also limit the ability to design studies to better understand the mechanisms underlying multivariate CEs and to assess, plan for, and mitigate the consequences of multivariate CEs.

Here, we present the first global climatology of different multivariate CEs consisting of two hazards co-occurring in space and time. We combine 12 different hazards from observations complemented with the ERA-Interim reanalysis (see "Methods") to form 27 hazard pairs with the potential to cause ecological and socioeconomic impacts. The possible impacts of some hazard pairs are more obvious than others. Supplementary Table 1 presents a list of possible ecological and socioeconomic impacts, including less obvious potentially impact-bearing hazard combinations. For example, the joint occurrence of low streamflow and HW may not immediately appear important, but can lead to increased transport costs due to shipping delays, and the requirement of additional refrigeration and storage[21,22]. Other combinations might cause a joint impact in the sense of monetary loss due to crop failure caused by HWs, and/or drought conditions paired with hail damage of crop and/or property in the same region. Using daily observations (where available) complemented by reanalysis data (Supplementary Table 2), we determine the annual and seasonal occurrence probability of these hazard pairs for the period between 1980 and 2014, and identify regional hotspots for the occurrence of multivariate CEs. Our results provide initial guidance of which multivariate CEs need to be included for risk assessments in particular regions. Our results also provide a dataset that can be used to assess the skill of climate models in simulating the occurrence of multivariate CEs. Combined with studies that examine whether climate models reproduce the driving mechanisms behind multivariate CEs correctly, our findings have the potential to identify those models best suited for predicting multivariate CEs in the future.

## Results

**Compound event hotspots.** In the following, CE hotspots are defined as geographical regions with short return periods in the joint occurrence of a specific hazard pair. In the multivariate context of CEs discussed in this paper, return periods (hereafter joint return periods; RP) are based on the probability that both hazards in a given pair exceed their individual threshold simultaneously (see "Methods"). The geographical joint occurrence of key hazard pairs (see list in Supplementary Table 1) is shown in Fig. 1 (relating to dry conditions) and Fig. 2 (relating to wet conditions; other hazard pairs are shown in Supplementary Fig. 1). Broadly, similar hazard pairs lead to similar regional hotspots. The occurrence for hazard pairs containing HW and dry conditions (low precipitation or a standardised precipitation index (SPI) below −1.3, hereafter lowP and drought) are located at midlatitudes (Fig. 1a, b). Hotspots for strong wind and drought CEs occur in isolated regions around the equatorial regions and at midlatitudes (Fig. 1c). Other hazard pairs (Fig. 1d, e) display strongly regional signatures (e.g., North America, eastern Europe, and Russia, Fig. 1d) or little clear regionality over most continents (e.g., Fig. 1f). In the case of wet hazard pairs (Fig. 2), eastern North America is a hotspot for the majority of multivariate CEs, e.g., high precipitation (highP) and hail, wind and hail, highP and high streamflow (highQ), and wind and highQ (Fig. 2a–d). This suggests a high susceptibility of this region to compound flooding and storm damage. CEs that involve extreme storm surges (hereafter surge) form hotspots along the western European coast (Fig. 2e–g) and both coasts of North America (Fig. 2e, g).

**Statistical dependence between hazards forming compound events.** The consequence of the statistical dependence between a hazard pair on its joint occurrence probability can be presented as a likelihood multiplication factor[7] (LMF). The LMF is the ratio of the observed empirical exceedance probability and the probability assuming independence between the hazards (Supplementary Figs. 2, 3 and 4). If the hotspots in the occurrence of one CE are reproduced in the global patterns of its corresponding LMF then the hazards making up the CE are likely strongly correlated due to their driving mechanisms, and are not the result of data coverage or baseline threshold choices. For example, the hotspots in highP and hail, wind and hail, highP and highQ, and high wind and highQ (Fig. 2a–d) located in North America coincide with high LMF values (Supplementary Fig. 3a–d). This suggests a common cause of hazards such as severe convective storms that can generate flash floods, with hail damage intensifying the risk of

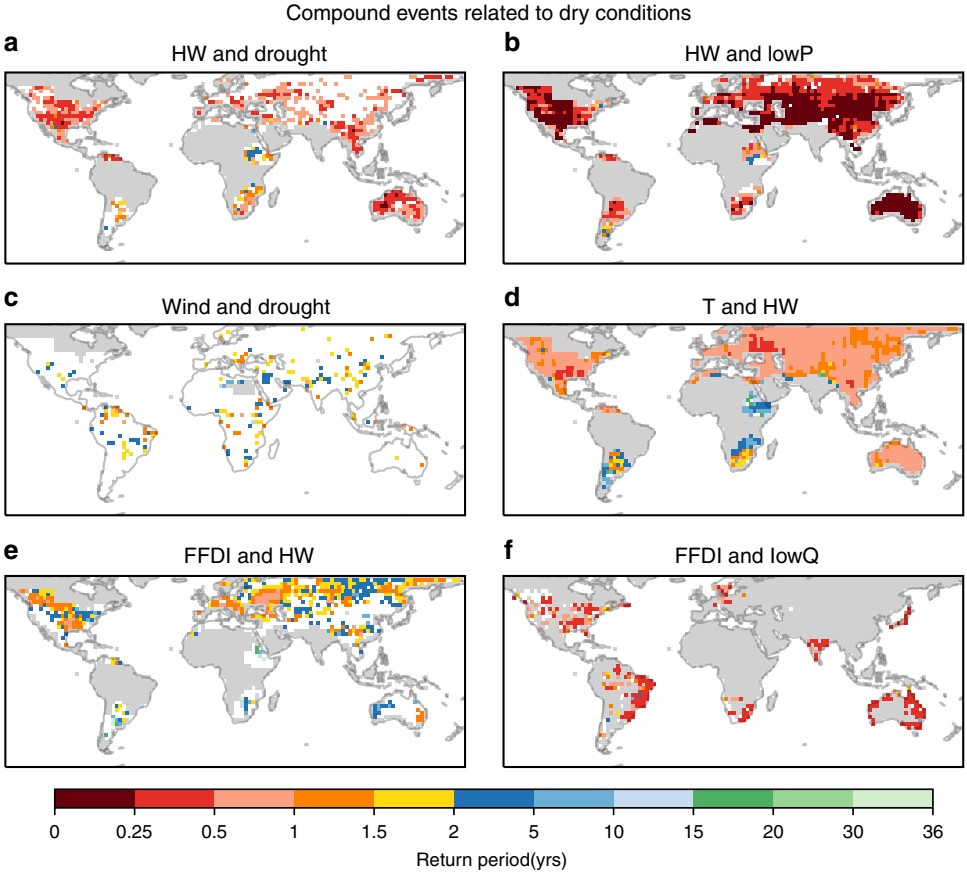

**Fig. 1 Hotspots of joint occurrence of different hazard pairs related to dry conditions.** This includes meteorological drought (drought) and hydrological drought (lowQ) in combination with heatwaves (HW), extreme temperature (T), McArthur forest fire danger index (FFDI), and low streamflow (lowQ). Shown are CEs consisting of **a** heatwave and meteorological drought, **b** heatwave and hydrological drought, **c** strong winds and meteorological drought, **d** high temperatures and heatwaves, **e** high fire danger and heatwave, and **f** high fire danger and hydrological drought. Joint occurrences are given as RP of the actual probabilities derived from the data allowing the direct comparison of different panels, and to Fig. 2 and Supplementary Fig. 1. Only statistically significant values are shown ($p ≤ 0.05$); statistically insignificant values are masked (white). Grey areas indicate regions without data coverage (Supplementary Fig. 10), or where percentile values of at least one hazard falls below the minimum required percentile value (Supplementary Table 2).

water damage. Similarly, the hotspots of CEs along the coast of Western Europe and the global hotspot of highP and surge in northeast Australia are accompanied by high LMF values (Fig. 2 and Supplementary Fig. 3). A possible common driver for these events are large-scale low-pressure systems[23]. We find clear hotspots for wind and highP (wind–highP) CEs along the northwest coasts of North America and Australia, the west coasts of Portugal and Madagascar, and the east coast of North America (Fig. 2h). In northwest Australia and eastern North America, this pattern resembles the footprint of landfalling tropical cyclones. At midlatitudes, the hotspots coincide with regions with a high occurrence frequency of atmospheric rivers, long filaments of increased water vapour transport that occur in relation to poleward moving extratropical cyclones[24,25]. Atmospheric rivers have been previously linked to the joint occurrence of wind–highP[24], as well as storm surge[24,26], highlighting the importance of these systems in driving wind–highP CEs. There has also been extensive research on tropical cyclones as drivers of high wind and/or precipitation extremes[4,27,28].

**Relative importance of compound events in different geographical regions.** We next focus on six continental regions (Supplementary Fig. 5) and examine the relative importance of different hazard pairs. Hazard pairs containing temperature and precipitation generally contribute to the majority of CEs (Fig. 3).

Coastal/hydrological and temperature-related CEs are relatively less important due to their low frequency and their strong seasonal link to summer, respectively (Supplementary Figs. 6 and 7). The role of hazard pairs varies strongly with region (Fig. 3). In North America, highP–highQ events are the dominant precipitation-related CE, while other CEs contribute <5% each (Fig. 3a), suggesting increased probability of flooding from CEs. Over Africa, the wind–drought CE exceeds 20%, suggesting a susceptibility to dust storms. In other regions, highP–highQ (South America and Oceania) and wind–drought (Europe and Asia) events are the most common. For precipitation and temperature-related CEs (Fig. 3b), McArthur forest fire danger index (FFDI) and drought, and lowP and HW are the most common in almost all regions.

**Seasonality of regionally important compound events.** We next examine the seasonality of CEs, noting that limited data precludes a reliable analysis of trends over the period 198–2014. Many precipitation- and temperature-related CEs occur mainly during spring, and the extended summer period (Fig. 4). For instance, in the northern hemisphere FFDI–drought and T–lowP CEs mainly occur during summer (May–August, Fig. 4b), potentially increasing fire danger and economic losses in important agricultural regions of the US and Europe. In the southern hemisphere, these CEs generally peak in spring, although the

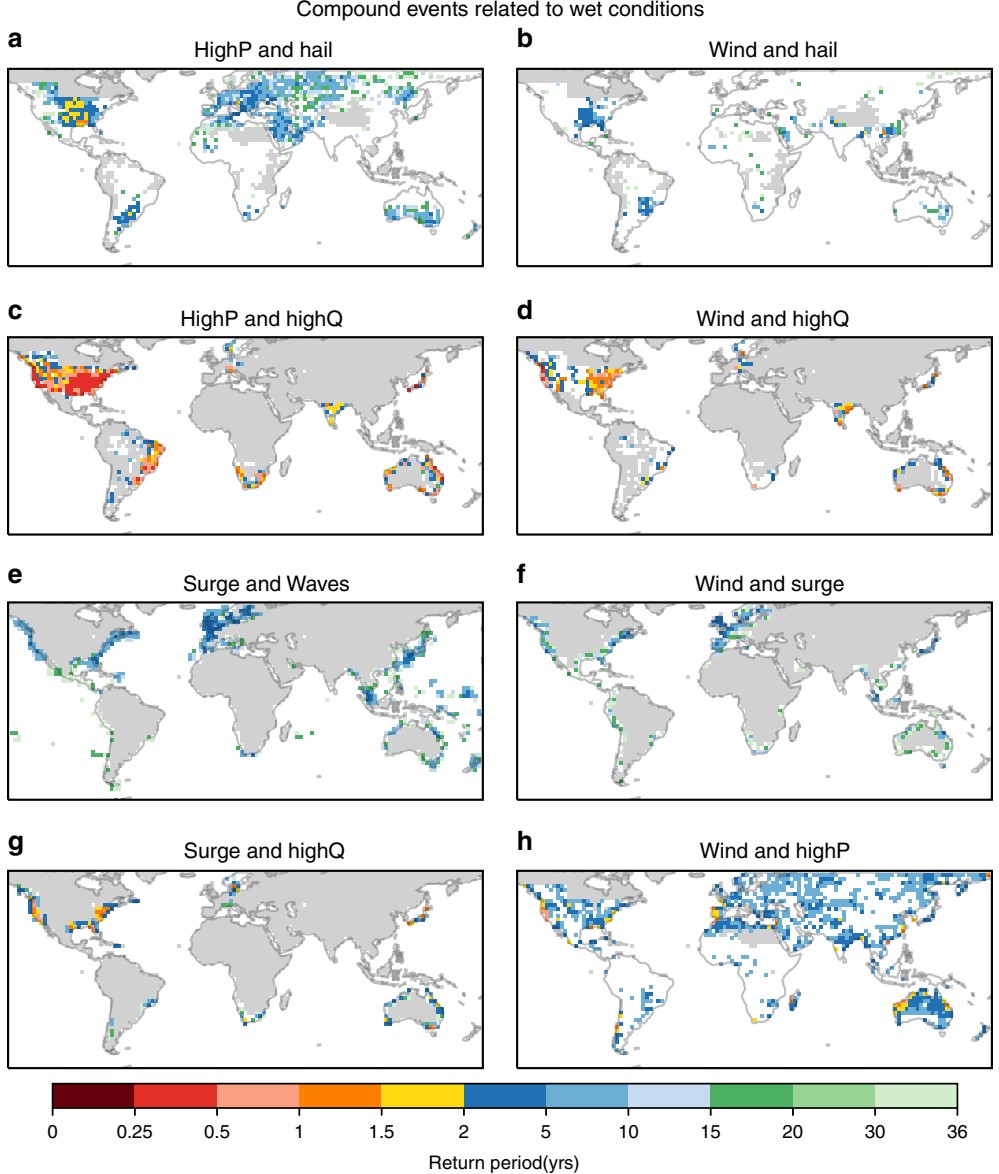

**Fig. 2 Hotspots of joint occurrence of different hazard pairs related to wet, potentially flood causing conditions.** Shown are combinations of high precipitation (highP), strong winds (wind), high probability of large hail (hail), high streamflow (highQ), and high storm surge (surge). Shown are CEs consisting of **a** high precipitation and strong wind, **b** strong winds and hail probability, **c** strong wind and highQ, **d** strong wind and highQ, **e** high surge and high waves, **f** strong winds and high surge, **g** high surge and highQ, and **h** strong winds and high precipitation. Joint occurrences are given as return period of the actual probabilities derived from the data allowing the direct comparison of different panels, and to Fig. 1 and Supplementary Fig. 1. Only statistically significant values are shown ($p \le 0.05$); statistically insignificant values are masked (white). Grey areas indicate regions without data coverage (Supplementary Fig. 10), or where percentile values of at least one hazard falls below the minimum required percentile value (Supplementary Table 2).

seasonality is more variable most likely due to the variety of climate zones included in the different regions. The peak in FFDI–drought CEs co-occurs with the peak in wind–drought CEs, potentially exacerbating fire danger. CEs consisting of HWs and lowP occur through the year in each region, but are generally more frequent in local summer or autumn except for Asia, where the peak season occurs in boreal winter. The seasonality of CEs reflects the seasonal occurrence of warm and dry conditions, resulting from the correlation between temperature and low precipitation, the key drivers for hazards in this CE group.

The dominant precipitation-related CEs show a more diverse seasonality. The highP–highQ CEs in North America mainly occur in late spring/early summer, while the season in Europe extends from late spring (May) to early winter (Dec; Fig. 4a). This overlap with the European storm season suggests that large-scale

low-pressure systems play a significant role in the occurrence of compound floods in Europe, while they play a lesser role in North America. In the southern hemisphere, highP–highQ CEs occur mainly in austral summer with some occurrences in early autumn in South America. This can be linked to the seasonal climatology of severe convective storms, and tropical cyclones at low latitudes.

**Drivers behind the occurrence of compound event hotspots.** The regional differences in hotspots and the seasonality of occurrences are caused by the drivers of the different CE types. As previously shown, CEs related to dry conditions are often located in inland areas particularly in North America, eastern Europe/ Russia, and to some extent Australia (Fig. 1b, c). They reflect the close link between temperature, precipitation/meteorological

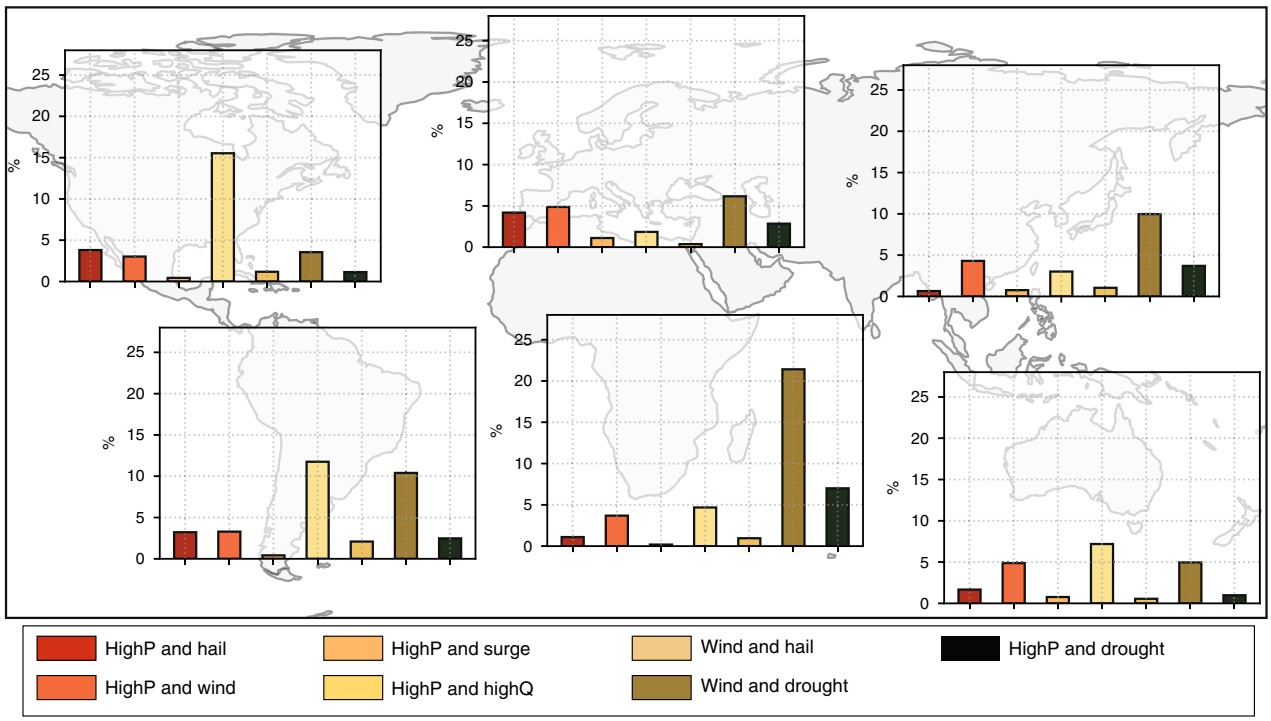

**a** Precipitation-related compound events

Legend:
- HighP and hail
- HighP and wind
- HighP and surge
- HighP and highQ
- Wind and hail
- Wind and drought
- HighP and drought

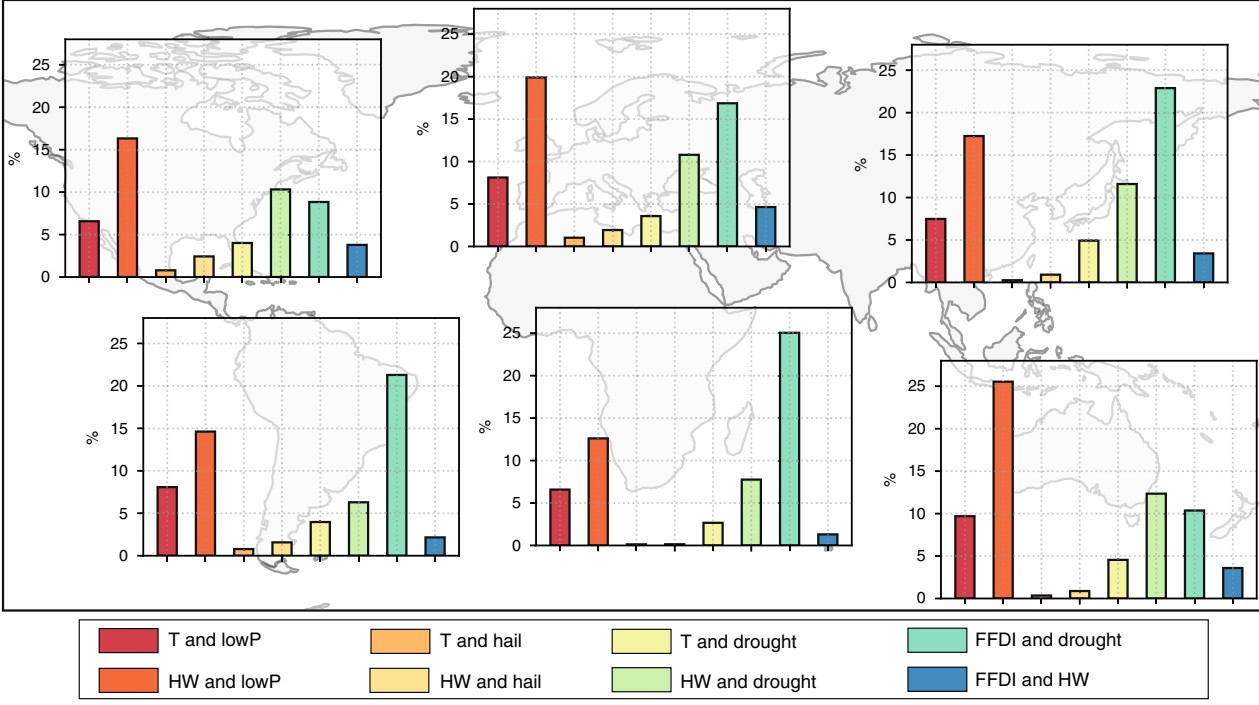

**b** Precipitation- and temperature-related compound events

Legend:
- T and lowP
- HW and lowP
- T and hail
- HW and hail
- T and drought
- HW and drought
- FFDI and drought
- FFDI and HW

**Fig. 3 Relative importance of hazard pairs per region.** Percentages are weighted by number of grid cells in the region taking into consideration the different data coverage of different hazard pairs (Supplementary Figs. 11–13). **a** Hazard pairs related to extreme high and low precipitation, including combinations of high precipitation (highP), high probability of large hail (hail), strong winds (wind), extreme storm surge (surge), high streamflow (highQ), and meteorological drought (drought). **b** Hazard pairs related to extreme precipitation and temperatures, including combinations of high temperatures (T), low precipitation (lowP), heatwaves (HW), high probability of large hail (hail), low SPI (drought), and extreme McArthur forest fire index (FFDI) values. Hazard pairs containing low streamflow (lowQ) were removed for consistency (univariate occurrence probability of lowQ inconsistent with other hazards due to threshold choice and gridding of station data; see "Methods" and Supplementary Fig. 9).

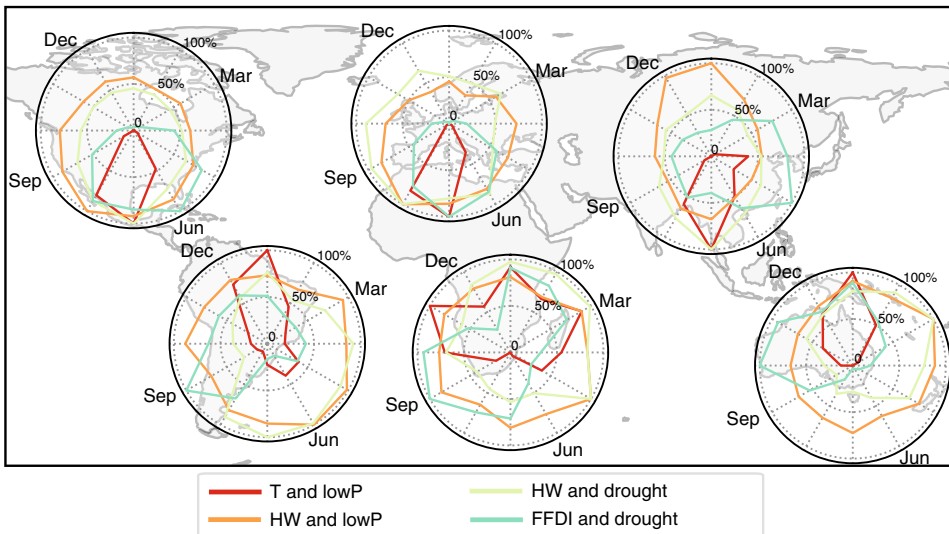

**Fig. 4 Seasonality of the most dominant multivariate CEs per region.** Values are relative to the total number of events in the month with the most occurrences. CEs are grouped into those containing **a** precipitation-related combinations, including the hazards high precipitation (highP), high probability of large hail (hail), strong winds (wind), high streamflow (highQ), and low SPI (drought); and **b** temperature and precipitation-related hazard pairs consisting of high temperatures (T), low precipitation (lowP), heatwaves (HW), meteorological drought (drought), and extreme McArthur forest fire index values (FFDI).

drought, and soil moisture[7]. Synoptic features facilitating these conditions include atmospheric blocking systems and other stable, long duration atmospheric features. In Southeast Asia and Australia, for example, global climate modes of variability, including the El Niño/Southern Oscillation and the Indian Ocean Dipole can cause weather conditions leading to dry CEs, which can prevail from months to years[29]. In contrast, CEs associated with wet conditions generally have a shorter duration ranging from hours to several days[30]. As such, their hotspots are often caused by atmospheric low-pressure and frontal systems[31,32]. For instance, eastern North America is a hotspot for highP and strong winds combined with hail and highQ[31]. These hazards are related to small- to meso-scale severe convective systems[33,34]. In contrast, Europe is a hotspot for CEs containing high surges, which require strong winds affecting a large area for an extended period of time, often associated with large-scale low-pressure systems, such as extratropical cyclones[32]. This is also reflected in the seasonality of

surge and wind-related CEs that coincide with the increased occurrence of storms in Europe during autumn and winter (Fig. 4 and Supplementary Fig. 4). An important factor contributing to the formation of hotspots of surge-related CEs in north-western Europe is the form of the North Sea, and the way surge waves travel through it[32]. These examples give an insight into the drivers and their regional characteristics. More thorough regional assessments are needed to determine the complex interplay of these drivers of regional hotspots and identify possible other contributing factors. However, the global pattern of multivariate CEs provides an immediate challenge for global climate models given the socioeconomic significance of these phenomena.

## Discussion

We present a first global climatology for a wide range of multivariate CEs with the potential to cause severe ecological and

socioeconomic impacts based on observations, and the ERA-Interim reanalysis product that spans three decades (1980–2014). We used reanalysis data to supplement observations; not all hazards could be derived from observational data due to the lack of available data products (e.g., wind speed and wind-related hazards, including storm surge and wave height). In many parts of the world, the reanalysis products are constrained by high-quality observations, but reanalyses are poorly constrained in data-sparse regions. This leads to unavoidable geographical variations in the reliability of our estimates of the joint probabilities of hazard pairs. We were careful in choosing the best available observational data with daily temporal resolution. We note our use of reanalyses was restricted to the ERA-Interim reanalysis product. We did not test the sensitivity of our results to alternative reanalyses for individual variables because taking one component of a CE (e.g., wind) from an alternative source would create physical inconsistencies with another component from ERA-Interim (e.g., storm surge). Thus, a repeat of our analysis with alternative observations and newer reanalyses is encouraged, but we recommend whole-scale replacement of the ERA-Interim when available.

While our global results are limited by data coverage, and uncertainties in both observational and reanalysis data, high-resolution regional datasets do exist that could be used to replace the global products. We therefore examined whether replacing the global datasets with alternative high-resolution data reproduce the main features of the global hotspots map for the most important CEs over Australia (Supplementary Fig. 8). Our results show a high degree of similarity in the overall patterns of CEs using alternative data, suggesting the regionally dependent hotspots identified in the global analysis, and the varying regional importance of different hazard pairs are robust. We therefore suggest that the global-scale analysis provide a useful first step in informing risk analysts and stakeholders concerned with identifying risk from multivariate CEs. While a translation of the CE hotspots identified in this study into risk maps would be useful for policy decision making, it falls outside of the scope of this study as this requires additional information about regional vulnerability and exposure in accordance with the IPCC framework[1,4]. The identified CEs, however, are an indicator for the possibility of adverse impacts. Our results are therefore an important first step and represent a foundation for the analysis of actual impacts caused by the different CEs assessed in this study. Further, the identification of hotspots highlights regions with the potential of increased risk, indicates where to focus future investigations of underlying mechanisms of co-occurring hazards, and provides a starting point for global impact analyses.

The class of multivariate CEs analysed in this study is only one of the possible types of CEs, all of which have potential to cause ecological and socioeconomic impacts. We selected multivariate CEs because this type has received the most attention in recent years[3] and a climatology of these events is therefore timely. Our decision to focus on CEs using a daily time step provides us with a rich sample to calculate the co-occurrence of pairs of hazards. We recognise that events with the duration of 1 day will not always cause significant socioeconomic impacts for all hazard combinations considered in this study. Adding a temporal dimension in the analysis is an area worthy of future work, but we note that it would increase the complexity quite substantially and reduce the data available for analysis.

We are confident that the analysis of multivariate CEs in this study, and the hotspots identified, are a necessary step forward for research into CEs. Our results provide (1) guidance on the occurrence probability of CEs, which can be considered as precursors of actual CE impacts, (2) information on the regional importance of CEs, and (3) a climatology to evaluate how well global climate models simulate CEs. Establishing those models

with skill in simulating CEs, and those models that capture the right statistics with the right physical mechanisms, would provide planners and risk analysts clarity on which climate models are best suited to explore changing risks associated with CEs under a changing climate.

## Methods

**Data.** The analysis uses a combination of observational and reanalysis at the daily timescale that has been gridded to a common $2.5° × 2.5°$ grid with 73 latitude bands and 144 longitude bands, spanning the period from 1980 to 2014 (12,874 days of record per grid cell). We consider a total of 12 different land-based hydroclimate variables and indices, namely daily high and low precipitation sums, SPI as a metric for meteorological drought, high and low streamflow, daily maximum temperatures, the excess heat factor (EHF) as a measure for HWs, the FFDI to quantify fire weather conditions, the probability of hail, maximum wind speeds, maximum storm surge, and maximum wave heights. The sources for the different hazards were chosen to ensure use of high-quality data, consistency of observation locations, and global coverage of the data, as far as possible. As noted earlier, to avoid physical inconsistencies, we use one reanalysis product for those variabilities not available from observational data on the global scale. Details on each dataset are listed below.

*Temperature and heatwaves.* Daily maximum temperatures were taken from the HadGHCND observational dataset[35]. HadGHCND is the largest available repository of global daily in situ observations for temperature. It has been specifically designed to analyse extremes that has been relied upon in past studies[36,37], which makes this dataset particularly suitable for this study.

The EHF index is calculated from HadGHCND using the method outlined by Perkins and Alexander[11]. HW events are identified as at least three consecutive days where the daily mean temperature exceeds the calendar day 90th percentile. This relative approach to define HWs captures events at all latitudes, including those events in colder climates and during local winter months. This ensures that we capture events which can have significant impacts due to system resilience and adaptation even if absolute temperatures might not be high.

*Precipitation and drought.* Daily precipitation sums were derived from the $1° × 1°$ observational dataset REGEN[38]. This dataset represents a combination of two of the largest in situ observational repositories, namely GPCC and HadGHCND. This reduces biases and uncertainties associated with single-source datasets, while incorporating information from the same observational system as the temperature dataset chosen here. As such REGEN is highly suitable for the analysis of CEs in the context of this study. From the REGEN precipitation data, monthly SPI values were calculated with the Climpact2 software (https://climpact-sci.org/get-started/), using the 3-monthly running means and subsequently transformed into daily data by assigning each day the monthly value.

*Other hazards.* Wind speed (ws) was calculated from the 3-hourly instantaneous 10-m zonal ($v$) and meridional ($u$) wind components of the ERA-Interim reanalysis provided by the European Centre for Medium-Range Weather Forecast[39] as ws = $\sqrt{u^2 + v^2}$. The final analysis uses the daily maximum of this 3-hourly wind speed data. The choice of using ERA-Interim was based on the availability of additional hazards calculated using this reanalysis product, such as probability of large hail, storm surge, and FFDI[40,41].

The streamflow data is taken from the Global Streamflow Indices and Metadata archive, which contains daily data from over 35,000 daily streamflow timeseries[33,42,43].

More detailed information about the different datasets can be found in the referenced literature (Supplementary Table 1).

*Interpolation and transformation of station data.* Bilinear interpolation was used to translate data from its original resolution to the 2.5° grid used here. Station data, namely surge height and river discharge, are transformed to grid data by assigning the stations to the relevant grid cells of the field data. Grid cells containing more than one station were assigned the value of the station with the highest surge height as derived from the Global Tidal and Storm surge Model (GTSM), using ERA-Interim 10-m wind components and sea level pressure[41]. In the case of river discharge, a grid cell is assumed to experience a high-flow (low-flow) extreme if any of the stations in that grid cell experience discharge values in the top 1% (lowest 10%) of the respective river basin. For grid cells with more than one streamflow station, this approach implies that the occurrence probability of high-flow (low-flow) events is increased, since there are multiple stations that have the potential to exceed the chosen threshold.

The handling of grid cells containing multiple stations in this way changes the RP of these three single hazards compared to the gridded hazards based on absolute variable values, e.g., temperature, wind, and heavy precipitation. Relative hazard indices, e.g., EHF and SPI, also differ from the globally homogenous RPs found for gridded hazards derived from absolute values. Global maps of RPs for the chosen thresholds and hazards can be found in Supplementary Fig. 9.

**Definition of compound events and hotspots**. We adopt the definition and explanation introduced by Zscheischler et al.[1,3], and define CEs as the combination of multiple drivers and/or hazards that contribute to societal or environmental risk. In this context, drivers include weather and/or climate processes, variables and phenomena that may span multiple spatial and temporal scales. Hazards are the immediate physical precursors to negative impacts. Within the wide range of CEs, we focus on multivariate CEs, that is, the co-occurrence of multiple climate drivers and/or hazards in the same geographical region[3].

Out of all the possible hazard pairs, we consider those with potential socioeconomic impacts and exclude those with no known socioeconomic effects (Supplementary Table 1). For each of the 27 identified hazard pairs, the two timeseries of the relevant climate variables/indices are combined. To identify potentially hazardous conditions caused by each hazard combination, a threshold is applied to both of its constituents. This is either a fixed value for the hazard indices EHF, SPI, hail probability, and FFDI, or a percentile threshold for climate variables. Percentiles are determined for each grid cell individually to account for regional differences (see Supplementary Table 2 for thresholds). Percentile values were derived using the Python NumPy function "percentile". This estimates the value of a specified percentile using provided data points (e.g., a value was associated with the 90th percentile if ~90% of the data points were equal to or less than that value). Linear interpolation was used to determine percentile values when they fell between two data points.

Results are presented as RPs, i.e., the inverse of occurrence probability which in this study is determined as follows. If both hazards in a pair jointly exceed their respective threshold on the same day in the same grid cell then the day is marked as a CE. Land-based hazards, storm surge, and wave height do not share the same grid cells, with the latter occurring only over the sea. To identify CEs formed by a combination of ocean-based and a land-based hazard, coastal grid cells are defined as land grid cells with at least one neighbouring ocean grid cell. As an example, for the combination of two land-based hazards, a day is considered a CE of the type wind and precipitation, if the total precipitation and the daily maximum wind speed on that day both lie within the top 1% of events during the study period. In terms of occurrence probability, this means that if the two hazard pairs are independent, the probability of their joint occurrence should be $0.01 \times 0.01 = 0.0001$, which is equivalent to approximately one occurrence during the study period (12,874 days).

To identify CEs containing surge and/or wave height each coastal cell is linked to exceedances in surge in the four surrounding cells (both ocean and land cells with land cells having no values). If the coastal grid cell exceeds the threshold of the relevant land-based hazard at the same time as the surge level in one of its four surrounding grid cells, then the grid cell is marked as having experienced a CE. This increases the region that we assume an extreme surge event can affect and changes the grid-cell-wise univariate RP for surge from a uniform ~100 days to lower values depending on station coverage of the GTSM dataset. The resulting RP is shown in Supplementary Fig. 9. Information about the coverage of different hazard pairs are shown in Supplementary Fig. 10.

We note this is one specific combination of bivariate probability, i.e. both hazard X and hazard Y: $P(X > x \wedge Y > y)$. However, multivariate risk assessment can incorporate more than multivariate CEs and depending on the impact one may consider a combination of different joint probability described by a variety of different possible hazard combinations[4], e.g., both hazard X and hazard Y: $P(X > x \wedge Y > y)$ and hazard X or hazard Y: $P(X > x \vee Y > y)$[44]. For the purpose of this paper we note this is not appropriate due to the decision to analyse multivariate CEs exclusively.

Global CE hotspots are determined by focusing on the identification of hotspots for each CE type/hazard pair individually. For this, we calculate the joint occurrence probability for each grid cell by adding the number of all joint exceedances in this grid cell and dividing it by the total number of days in the study period. The results are then presented as RP by dividing the inverse of the probability by the number of days per year. For each hazard pair X and Y, we use the global binary maps that show when (time) and where (latitude and longitude) each hazard exceeds its respective threshold. That is, we have two $12784 \times 73 \times 144$ matrixes **X** and **Y**, whose elements are either 0 (hazard does not exceed the threshold) or 1 (hazard exceeds the threshold). Bold capital letters indicate that these are matrixes. Matrix elements are displayed in italic capital letters with the subscript "$i$".

We then compare where the same elements in both **X** and **Y** are equal 1, i.e., where both hazards exceed their threshold at the same time and latitude–longitude point. This results in a three-dimensional matrix **Z** ($12,784 \times 73 \times 144$) of which each element ($Z_i$) was assigned a value of either 0 or 1 following the below rules:

$$Z_i = \begin{cases} 1, & (X_i = 1 \wedge Y_i = 1) \\ 0, & \text{otherwise} \end{cases}. \tag{1}$$

The probability map of joint exceedance ($P(X \wedge Y)$) is then the sum of **Z** over time (**W**), divided by the number of days in the study period ($n$days):

$$P(X \wedge Y) = \frac{\sum_t \mathbf{Z}(t, \text{latitude}, \text{longitude})}{n\text{days}} = \frac{\mathbf{W}(\text{latitude}, \text{longitude})}{n\text{days}}. \tag{2}$$

As such, matrix **W** with the dimension $73 \times 144$, i.e., latitude × longitude, represents a map with the total numbers of joint exceedances of hazards X and Y throughout the study period. The return period (RP with [RP] = years) is then

calculated as:

$$\text{RP}(X \wedge Y) = \frac{1}{P(X \wedge Y) \times 365}. \tag{3}$$

**Likelihood multiplication factor**. We use the LMF[7] to illustrate the impact of the possible correlation between hazard pairs on their joint occurrence probability. The LMF is the ratio of the actually observed probability of joint occurrence ($P_{\text{actual}}$) and the probability assuming complete independence between the hazard pair ($P_{\text{indep}}$). Therefore, considering LMF as a matrix of dimension latitude × longitude, each element of LMF is calculated as $\text{LMF} = \frac{P_{\text{actual}}}{P_{\text{indep}}}$. The LMF for each hazard pair, e.g., hazards X and Y, is determined for each grid cell individually by calculating the grid cell-specific $P_{\text{indep},i}$ and $P_{\text{actual},i}$ as:

$$P_{\text{indep},i} = P_i(\text{hazard } X) \times P_i(\text{hazard } Y) = \frac{\sum_t X_i}{n\text{days}} \times \frac{\sum_t Y_i}{n\text{days}}, \tag{4}$$

$$P_{\text{actual},i} = P_i(X \wedge Y) = \frac{W_i}{n\text{days}}. \tag{5}$$

The LMF varies between 0 and infinity, i.e., $\text{LMF} \in [0, \infty)$. If the two hazards are independent the LMF equals 1. For positively correlated hazard pairs the LMF is larger than one and increasing with the strength of correlation. For negatively correlated hazards the LMF falls between 0 and 1.

$$\text{LMF} := \begin{cases} <1 & \text{negative correlation} \\ = 1 & \text{independence} \\ >1 & \text{positive correlation}. \end{cases} \tag{6}$$

**Regional and seasonal analysis**. The regional analysis takes into account the different coverage of hazard pairs. As such, we first calculate the total possible number of events in this region to determine the relative contribution of a specific CE type in one region. For this, we multiply the number of grid cells covered by each individual CE by the total number of days in the study period. The total amount of possible CE occurrences per region is then the sum over the products of all individual CEs. This approach eradicates biases caused by the different coverage of the individual data products. Further, it ensures that coastal hazards are weighted in the same way as land-based hazards, which naturally can occur in a larger number of grid cells. It is important to note that the inhomogeneous distribution of streamflow stations globally causes some grid cells to contain more than one station (as mentioned above). The occurrence probability of a low-flow events in single-station grid cells is equal to that of a single station, i.e., 0.1%. In contrast, the occurrence probability of low-flow events in a multi-station grid cell is increased, since there are multiple stations that have the potential to experience low-flow conditions. In addition, the choice of the maximum threshold for low-flow events as values below the tenth percentile of the distribution is not proportional to that of the other hazards, for which values above the 99th percentile are considered as extreme. As a result, low-flow events are ten times more likely than the other hazards. This threshold choice for low-flow events is defensible on two grounds. First, negative impacts of low streamflow can start to occur for streamflow values below the tenth percentile. Second, this threshold has been used by the Copernicus Climate Change Service for their report on the state of the European climate (https://climate.copernicus.eu/river-discharge). The combination of these two special characteristics around this hazard artificially increases the importance of CEs containing low streamflow compared to the other CE types assessed in this study (Supplementary Fig. 9). To avoid another scaling factor that might introduce further uncertainty, the relative contributions of all CEs to regional occurrence only considers hazard pairs without low streamflow.

For the seasonal analysis, we add the number of joint exceedances from 1980–2014 for each month separately. The month with the highest number of events is used as a baseline to scale the number of events in the other months. The monthly sums are then displayed in a polar plot.

**Statistical significance assessment**. A significance test is applied for each hazard pair to assess whether the joint exceedance of a hazard pair is significant at a specific grid cell. We test the null hypothesis that the joint RP of the hazard pair found in each grid cell can be reproduced by chance and does not require any physical correlation between the two hazards. For this, we removed the physical correlation between the two hazards using a bootstrapping procedure and compared the resampled (1000 datasets) and observed timeseries. A result is considered statistically significant if the number of observed events is higher than 95% of the resampled timeseries. This is equivalent to a $p$ value of 0.05 or below. We perform the following steps, using capital letters in bold font to indicate matrixes, italic capital letters with the subscript "$i$" to represent matrix elements, and "*" to indicate resampled datasets:

*Step 1*: Without changing the dimensions of latitude and longitude, we use a bootstrapping method (with replacement) on the time axis of matrix **Y**, to generate 1000 alternate timeseries $\mathbf{Y}_n^*$ (with $n = \{\mathbb{N} | n \in [1, 1000]\}$). The time sequence of the exceedance matrix **X** remains unchanged in this step (i.e., observed occurrence of hazard X was used).

*Step 2*: We constructed a three-dimension matrix $Z_n^*$ ($12784 \times 73 \times 144$) following the same procedure that was applied to identify the observed joint occurrence of CE X–Y (matrix $Z$). Specifically, each element $Z_{n,i}^*$ was assigned a value of 1 if the same elements in both $X$ ($X_i$) and $Y_n^*$($Y_{n,i}^*$) are equal to 1. Otherwise, a value of 0 is assigned to $Z_{n,i}^*$. The Boolean matrix $Z_n^*$ ($12784 \times 73 \times 144$) represents the joint occurrence in space between hazard $X$ and hazard $Y$ that introduced by random chance over the analysis period (as the temporal association between hazard $X$ and hazard $Y$ was removed).

*Step 3*: We then sum each $Z_n^*$ over time to get the $73 \times 144$ matrixes $W_n^*$ (Eq. (7)) that represent the joint exceedance over space between hazard $X$ and hazard $Y$ that are introduced by random chance (i.e., a resampled replication of matrix $W$ used in Eq. (2)).

$$W_n^* = \sum_t Z_n^*(t, n\text{latitude}, n\text{longitude}). \quad (7)$$

Since $W_n^*$ contains discrete values, i.e., $W_{n,i}^* \in \mathbb{N}$, it is necessary to introduce artificial noise to each element of the $W_n^*$ matrixes to ensure that the probability of the null hypothesis, in this case 5%, remains unchanged. This is done by adding a matrix of the same rank as $W_n^*$ consisting of random elements ranging between $-0.0009$ and $0.0009$ ($N_n$ with $N_{n,I}$ $= \left\{ \mathbb{R} | N_{n,i} \in [-0.0001, 0.0001] \right\}$ and $n = \{ \mathbb{N} | n \in [1, 1000] \}$ to each $W_n^*$ so that

$$W_n^{*\prime} = W_n^* + N_n, \quad (8)$$

with (′) indicating that $W_n^{*\prime}$ is a perturbed version of matrix $W_n^*$. If the number of joint occurrences in <5% of the resampled realisations ($W_n^{*\prime}$) are higher than that in $W$, there is evidence to reject the null hypothesis that the joint exceedance between hazard $X$ and hazard $Y$ was introduced by random chance.

We also tested for field significance of the joint exceedance probability adopting a resampling approach to account for both spatial and temporal dependencies[45]. For this, we counted the total number of statistically significant grid cells ($n_{\text{sign}}$) in $W$ from Eq. (2). We then determined $n_{\text{sign}}$ for all $W_n^{*\prime}$ by comparing each of the resampled datasets $W_m^{*\prime}$(with $m = \{ N | m \in [1, 1000] \}$) with the other 999 replicas $\left( W_k^{*\prime}, \text{with } k = \{ N | k \in [1, 1000] \wedge k \neq m \} \right)$ and the original time matrix ($W$), using the same 5% threshold for significance as for $W$. The results presented in Figs. 1 and 2 are considered "field significant" if $n_{\text{sign}}$ in the original dataset is greater than in 97.5% of the resampled replicates $W_k^{*\prime}$ (Supplementary Fig. 11).

**Reporting summary**. Further information on research design is available in the Nature Research Reporting Summary linked to this article.

## Data availability
The storm surge data can be obtained online from the 4TU.ResearchData archive (https://data.4tu.nl/repository/uuid:29614991-345e-4ffd-be22-2930912a2798). The applied fire weather indices (FFDI) data were provided through the Zenodo service (https://zenodo.org/record/3251000#.XcyoXzIzaL4) and the daily large hail probability can be downloaded from the information system PANGEA (https://doi.pangaea.de/10.1594/PANGAEA.888881?format=html#download). This study used gridded daily temperature data from the Met Office Hadley Centre observations dataset available at https://www.metoffice.gov.uk/hadobs/hadghcnd/download.html. The precipitation data used in this study (REGEN) has been published with unique Digital Object Identifiers (DOIs) https://doi.org/10.25914/5b9fa55a8298c and is available via the Research Data Australia (RDA) web page https://researchdata.ands.org.au/search/#!/slug=rainfall-estimates-gridded-station-v10.

## Code availability
Any relevant code necessary to reproduce the results presented here is available upon request.

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

## Acknowledgements

The research was funded by the Australian Research Council Centre of Excellence for Climate Extremes (CE170100023) and was supported in part by the New South Wales Department of Planning, Industry and Environment. H.X.D. is currently funded by School for Environment and Sustainability, University of Michigan (U064474). J.Z. acknowledges funding from the Swiss National Science Foundation (Ambizione grant 179876). N.N.R. and J.Z. acknowledge the European COST Action DAMOCLES (CA17109).

## Author contributions

N.N.R. designed the research, conducted all analyses, and wrote the first draft of the manuscript. N.N.R., A.J.P., J.P.E., and A.U. contributed to the design and selection of the final figures. N.N.R., A.J.P., J.Z., S.W., and J.P.E. defined details of the framework for this study. M.B. provided an assessment of different precipitation datasets that was essential for the final decision to use the REGEN dataset. H.X.D. supplied the streamflow data. A.L.H. calculated the HW measure. A.D.L. performed the wind speed calculations. All authors contributed to data interpretation, analysis methods, and writing of the final manuscript.

## Competing interests

The authors declare no competing interests.
