## [Peer Review File · Nature Communications]

Reviewers' Comments:

Reviewer #1:

Remarks to the Author:

Review of "Global hotspots for the occurrence of compound events"

The authors provide a global analysis of the hazard of a range of 2-dimensional compound events (CEs). The study is comprehensive and thus very useful for a broader community. I do have some major comments though, and the study requires a major revision. I also have some doubts whether the study meets the requirements for publication in Nature Communications. In particular, the authors have a tendency to oversell the relevance of this work, as I will discuss below.

First, the authors define CEs as high-impact events, but they do not address these impacts in their study. This limitation is common in the study of compound events (see, e.g., the discussion in Bevacqua et al., 2017), and it has several consequences I will discuss below. Thus, the authors should make explicit that they are studying the potential of compound events rather than the events themselves. In a similar situation, Bevacqua et al. (2019) define *potential* compound flooding. I suggest to adopt this concept and refer to the mentioned study.

One limitation of not analysing the impact is the fact that no impact might occur as a consequence of the defined events at all. One reason is that some impacts occur only under specific geographical conditions. E.g., wind and high streamflow may require a specific wind direction and topography to cause an impact. Another reason is that some hazards might be too weak to cause an impact (see also below under the specific points). For instance, a heatwave defined using a relative threshold might - in a cold climate - be so cold that it does not cause major impacts. Another reason is related to time scales (see following discussion). Thus, the actual regions of CE hotspots will be much smaller than those identified. Again, this should be made transparent.

An issue which has also not been discussed in this context is the role of time scales. The authors consider daily events of, e.g., high temperatures and low stream flow. I doubt that such an event of a single day length would really cause a severe impact. The authors should discuss the relevance of their selected hazards in more detail and really clarify their impact relevance.

For other hazards the relevance of the compoundness is not really clear to me. For instance, I am not sure what is so relevant about co-occurring heatwaves and large hail events. The impacts are, to my understanding, quite different, such that no compound impact may be caused such as in the case of compound flooding or heat and drought. If there is such an impact, it should be explained better.

An issue which needs to be made transparent is the fact that the study is not purely observational. E.g., winds and hail are taken from reanalysis data. As long as a sufficient amount of observational data has been assimilated, this is not necessarily a problem. But for hail and similar variables, the output is a pure model product even based on parameterisations (such as cloud microphysics). Thus these data cannot be used as an independent observational source to evaluate models. This shortcoming needs to be made explicit and transparent in a prominent place (e.g., the introduction). Otherwise it just corroborates the impression that the authors oversell the relevance of their study.

The writing of the manuscript has been somewhat sloppy and needs substantial improvement, see also the specific comments. Regarding structure, the authors need to add a link between the general introduction and the results part. Here the main concepts (e.g., how a

hotspot is defined, what type of return period has been chosen) need to be at least briefly presented, with explicit links to the relevant methods sections. Otherwise the reader is lost.

The usage of the term "risk" is not very clear. The authors state (l 249) "Risk is defined as the probability of hazards and represents a convolution of the respective distributions of probability and consequences.". Isn't this self contradictory, as the hazard is only defined by a probability? A more useful definition, also in line with the IPCC framework (and I think this is what the authors mean) would be to define risk as stated in the last sentence, but then it is not the probability of a hazard, but relates to the impact.

Overall the study is, as stated above, very comprehensive and definitely useful for the broad readership of Nature Communications. As discussed, it has important limitations because impacts have not been directly considered. This is not the author's responsibility, almost all large-scale studies on compound events suffer from this shortcoming, simply because only few suitable large-scale impact observations exist. But this limitation should be made transparent. In many places, I have the feeling the authors oversell the relevance of their study. In fact, I am wondering whether the study merits publication in Nature Communications. The authors have made a big effort to collect data and to calculate compound hazards. But the study lacks originality beyond rather the straight forward analyses.

Specific Issues

37 does "generate" really refer to "interaction", as the grammar suggests, or rather to the "drivers" (then an "s" is missing)

46 I do not fully agree with this statement. If one had an impact time series, one could well do a univariate analysis as well. Only to understand the interplay, a multivariate analysis would be required, or in case no impact data was available (i.e., in a prediction context). The authors rather want to point out that the co-occurrence of two events is underestimated if the dependence is not modelled, but this is a different thing, and the authors should explain that precisely.

54 Also study 17 should be mentioned as a large-scale study of compound flooding - all the European coastlines have been investigated.

59 The authors slightly oversell the relevance of their study here - why does the lack of similar studies prevent the development of a better understanding of the mechanisms underlying compound events? I really don't buy that. Mechanisms causing, e.g., compound flooding or dry heat waves are in principle well understood. What is new is just the quantification.

69 Delete the "robust" - it is not needed here.

71 how does the dataset help to study the mechanisms behind CEs? I find this a bit far-stretched. The same holds even more for assessing the reliability of climate models to predict future changes in CE occurrence. This is important, but it requires process-based analyses.

75 there is no unique definition of return periods for the multivariate case (see e.g., Serinaldi 2015, or the discussion in Bevacqua et al., 2019). In the methods section, the type is specified (AND), but here should be a reference, and somewhere the choice should be explained, as it affects the results.

244 this section is not well organised. E.g., return periods are used implicitly throughout the section, but they are only defined very far to the end. The discussion of land-based hazards and storm surges appears just out of nowhere - is this the best place to put it?

258 are the percentile-based indices really relevant? I have been struggling myself here, but maybe the authors could help the reader a bit by giving some advise on the limitations of this approach. Consider a "heatwave" with temperatures above 20C in a cool climate - such a heatwave will cause very different impacts - if at all! - than a heatwave of 40C in a hot climate. This further highlights the need to discuss the "potentiality" character of your study.

360 This section is very sloppy. For a significance test, you need to clearly define a null hypothesis. What is the null hypothesis here?

491 Reference 17 should be updated (see below in the list of references)

Supplementary Information:

The preparation of the supplementary information has been very sloppy. No author list is provided, and the page breaks are sometimes annoying. E.g., Table 1 ends just after the break, so that it is not quite easy to find the caption.

References

E. Bevacqua, D. Maraun, I. Hobaeck Haff, M. Widmann and M. Vrac: Multivariate statistical modelling of compound events via pair-copula constructions: analysis of floods in Ravenna (Italy), *Hydrol. Earth Syst. Sci.*, 21, 2701–2723, 2017.

E. Bevacqua, D. Maraun, M.I. Vousdoukas, E. Voukouvalas, M. Vrac, L. Mentaschi, and M. Widmann: Higher potential compound flood risk in Northern Europe under anthropogenic climate change, *Science Advances* 5, eaaw5531, 2019.

F. Serinaldi: Dismissing return periods! *Stoch. Environ. Res. Risk Assess.* 29, 1179–1189, 2015.

Reviewer #2:

Remarks to the Author:

Review of NCOMMS-20-09682-T, Global hotspots for the occurrence of compound events, by Ridder et al.

Ridder and colleagues present a global climatology of compound extremes by analyzing a set of combined climate variables and hazards co-occurring simultaneously. The analysis is an important first step and I'm generally supportive of the manuscript. The challenge is the amount of information presented in the figures and the supplement, and the fundamental lack of space in a letter format to give the results the due diligence they deserve given the work these authors have done. I leave it to the editor to decide whether this is an appropriate format, but I note that there are 16 supplementary figures, which strikes me as a lot. Regardless of that, I have four principal comments along methods and CE philosophy that the authors might consider as they revise the manuscript.

Methodology:

1. Data uncertainty: The authors have done a nice job corralling a lot of data and being thoughtful about the compound pairs they analyze. One practice that is emerging however, particularly when estimating at the tails of a joint distributions in observations and reanalyses, is the use of multiple datasets to provide an observational ensemble (e.g., Horton et al., *Nature* 2015; Coffel et al. *Earth's Future* 2019). Given the rigor of some of the analyses (e.g., field significance, which needs to be employed much more in climate studies), I was surprised the authors were content with using a single data source for each variable, especially given the fine temporal resolution of their analysis and their emphasis on building the first climatology of the co-occurrence of extremes. This sampling uncertainty can be quite large and while the bootstrapping addresses some portion of this uncertainty, it rests on the assumption that the sample is representative of the population, which it may not be. I would encourage the authors to include more observational and reanalysis datasets in their analysis.

2. Percentile estimation: For those climate variables where percentile estimates are used, how are those estimated? There is no information provided on the distribution fitting methods or parametric assumptions, which is another major source of uncertainty not included.

CE philosophy:

3. Definitional: I recognize one of the co-authors conceived of the definition of compound extremes as either variables and/or hazards. But I wondered about the focus on only those that were simultaneous. Certainly a compound pair would be impactful if they were back to back?

4. Impact: My previous comment, I think, links to one of the main things that is missing from the analysis, which is the actual impact. The analysis assumes that these percentiles or occurrences are themselves impactful to people and their socio-economic systems. The actual impacts of these things are divorced from the analysis presented, which is really problematic from an interpretation standpoint. A hotspot may emerge as a hotspot by this analysis, but there may not be any actual consequences of being a hotspot—meaning the compound pair does not actually have socio-economic impacts. If the point is to aid risk assessment, then a glaring omission is the assumption that a top-down approach like this is actually meaningful in informing risk, without some assessment of how often such CEs actually select for on-the-ground impacts. I would encourage the authors to think about how their extreme occurrence relates to outcomes they seek to inform the risk management of. Consider the recent work of Kornhuber et al. *Nat Clim Chg* 2020, where they linked quasi-stationary wave occurrence to actual agricultural impacts. The scientific community, if it is actually interested in informing risk assessment, must move beyond assuming that a 90th percentile heat event is problematic and instead show that it is problematic.

We thank both reviewers for their advice, detailed comments and suggestions. We have addressed each of their comments carefully and adjusted the manuscript accordingly. Below you will find our response to each reviewer comment. The reviewer's remarks are in black, **our response** is highlighted in blue and **extracts from the manuscript** are in red, with **new text that has been added** marked in bold.

Kind regards,

Nina Ridder (on behalf of all authors)

REVIEWER COMMENTS

Review of "Global hotspots for the occurrence of compound events"

The authors provide a global analysis of the hazard of a range of 2-dimensional compound events (CEs). The study is comprehensive and thus very useful for a broader community. I do have some major comments though, and the study requires a major revision. I also have some doubts whether the study meets the requirements for publication in Nature Communications. In particular, the authors have a tendency to oversell the relevance of this work, as I will discuss below.

We thank the reviewer for bringing this to our attention. We made sure to adjust the text to not overstate the contribution our research brings. Examples for this are throughout the manuscript. Many are the addition of a word or phrase, but in total we think address the reviewer's comment appropriately to avoid overselling our work.

Introduction

"This lack of a **global** geographical fingerprint of CEs limits the ability **to design studies** to better understand the mechanisms underlying CEs and to assess, plan for and mitigate the consequences of CEs."

and

"Our results provide **initial** guidance of which CEs need to be included for risk assessments in particular regions. Our results also provide a dataset **that can be used** to assess the skill of climate models in simulating the occurrence of CEs. **Combined with studies that examine whether climate models reproduce the driving mechanisms behind them CEs correctly, the findings have the potential to identify those models best suitable for predicting CEs in the future.**"

Discussion

"[...] **A direct translation of the CE hotspots identified in this study into impact maps is however not possible due to the differentiation between risk and impact and the here applied definition of compound events.² In the IPCC framework, which we adopt for this study, risk is defined as "the potential for adverse consequences of a climate-related hazard" and results from the combination of hazard, vulnerability and exposure.^{2,4} Impacts are the consequence of risk without being interchangeable with risk. As such, while CEs as weather and climate hazards contribute to risk, they can occur without resulting in an impact. Their occurrence probability, however, is an indicator for the possibility of adverse impacts. Our results are therefore an important first step and represent a foundation for the**

analysis of actual impacts caused by the different CEs assessed in this study. The generation of impact maps are however beyond the scope of this study. Nevertheless, the identification of hotspots highlights regions with the potential of increased risk, indicates where to focus future investigations of underlying mechanisms of co-occurring hazards and provides a starting point for global impact analyses.”

“We are confident that multivariate CEs analysed in this study, and the hotspots identified, is a necessary step forward for research into CEs. For some, our results provide guidance on the joint probability of drivers of CEs and can be considered precursors of actual compound events. For others, our results provide guidance on CEs and a climatology to evaluate how well global climate models simulate CEs. ~~Global climate models used for future projections have been thoroughly evaluated for the mean state, variability, trends and their capacity to simulate extreme events.~~³⁵ Climate models can be evaluated against our climatology, and examined to test whether they can capture the patterns and statistics of the CEs presented here. Establishing those models with skill in simulating CEs **and capturing the correct statistics with the right physical mechanisms**, would provide policymakers, planners and risk analysts **clarity on which climate models are best suited to** explore changing risks associated with CEs under a changing climate.”

First, the authors define CEs as high-impact events, but they do not address these impacts in their study. This limitation is common in the study of compound events (see, e.g., the discussion in Bevacqua et al., 2017), and it has several consequences I will discuss below. Thus, the authors should make explicit that they are studying the potential of compound events rather than the events themselves. In a similar situation, Bevacqua et al. (2019) define *potential* compound flooding. I suggest to adopt this concept and refer to the mentioned study.

We thank the reviewer for highlighting that we need to clarify the definition and interpretation of terms used in the manuscript, particularly the term “Compound Event”. We appreciate that for some applications our interpretation that CEs do not necessarily lead to an impact might not be practical. However, in the more general context of this study, we do not agree with the suggested wording and the thus implied consequences of talking about “*potential* compound events” for the following reasons. The definition of compound events used in this study is: “*Compound weather and climate events refer to the combination of multiple drivers and/or hazards that contributes to societal or environmental risk.*” Following the IPCC framework on which this definition was based, (extreme) weather and climate events are defined as weather/climate variables exceeding a set threshold and are considered hazards, i.e. events that may cause negative ecological and/or socio-economic repercussions. Risk is considered the combination of hazard, vulnerability and exposure. Impacts are defined as the consequence of risk but are not interchangeable with the term risk. As such events can occur independently of whether or not the exceedance of a threshold causes an impact while still influencing risk.

However, we clearly need to address the issues that led to the Reviewer’s comments. To clarify this distinction, we adjusted the manuscript where necessary. We adjusted the first sentence in the Summary to read:

“Compound events (CEs) are weather and climate events that result from multiple hazards or drivers **with the potential to cause severe socio-economic impacts.**”

Further we changed the first paragraph of the Introduction to:

“Extreme weather and climate events often result from a combination of multiple hazards or drivers. These events are commonly referred to as Compound Events (CEs).^{1, 2, 3, 4} The interaction of multiple hazards and/or drivers that generates CEs often leads to more severe **ecologically and socio-economically damaging events compared to single-hazard events.⁵ **Multivariate risk assessment sometimes requires multivariate extremes to have an impact to be considered a CE. Here we use the risk framework of the Intergovernmental Panel for Climate Change (IPCC) in which extreme weather and climate events are considered a hazard based solely on the variable’s value compared to its distribution noting that rare meteorological phenomenon may not necessarily have consequences⁴.”****

“[...] high fuel load and strong winds. Adopting the IPCC’s definition of extreme climate events means that not all occurrences of a CE lead to an impact as these are dependent on a combination of hazard occurrence as well as vulnerability and exposure of the affected region/system. Our analysis therefore focusses on the occurrence of CEs defined as the joint probability of two hazards and we do not explore whether the CEs necessarily lead to impacts. For multivariate risk assessments that require CEs to have impacts, our results can be interpreted as the climatology of the precursors to CEs.”

We also added the following statement to the Discussion and Conclusions:

“[...] the varying regional importance of different hazard pairs provides a foundation to inform policy and stakeholders to mitigate impacts from multivariate CEs. A direct translation of the CE hotspots identified in this study into impact maps is however not possible due to the differentiation between risk and impact and the definition of compound events applied here². In the IPCC framework, which we adopt for this study, risk is defined as “the potential for adverse consequences of a climate-related hazard” and results from the combination of hazard, vulnerability and exposure^{2, 4}. Impacts are the consequence of risk without being interchangeable with risk. As such, while CEs as weather and climate hazards contribute to risk, they can occur without resulting in an impact. Their occurrence probability, however, is an indicator for the possibility of adverse impacts.“

And later in the same section:

“For some, our results provide guidance on the joint probability of drivers of CEs and can be considered precursors of actual compound events. For others, our results provide guidance on CEs and a climatology to evaluate how well global climate models simulate CEs.”

One limitation of not analysing the impact is the fact that no impact might occur as a consequence of the defined events at all. One reason is that some impacts occur only under specific geographical conditions. E.g., wind and high streamflow may require a specific wind direction and topography to cause an impact. Another reason is that some hazards might be too weak to cause an impact (see also below under the specific points). For instance, a heatwave defined using a relative threshold might - in a cold climate - be so cold that it does not cause major impacts. Another reason is related to time scales (see following discussion). Thus, the actual regions of CE hotspots will be much smaller than those identified. Again, this should be made transparent.

We thank the reviewer for pointing out that we need to clarify that with the occurrence of CEs we only analyse the potential for an impact (refer to our response to the previous comment). We do however disagree that a relative threshold to define a hazard is not suitable for this purpose. Most

systems are adapted to the prevailing climatological mean conditions. As a result, the system is put under varying degrees of stress with every anomaly, with higher stress caused by larger anomaly. As such, anomalously high temperatures in cold climates can cause severe impacts. Polar animals have a very different threshold for heatstroke than animals in the African Sahara for instance. Winter heatwaves with low absolute temperatures can also have detrimental effects in these ecosystems e.g. due to snow and/or permafrost melt. To avoid confusion when it comes to how we determined the occurrence of a heatwave (at least three consecutive days with a daily mean temperature exceeding the climatological calendar day 90th percentile), we added the following clarification to the methods section:

“The EHF index is calculated from HadGHCND using the method outlined by Perkins and Alexander (2013)¹¹. Heatwave (HW) events are identified as at least three consecutive days where the daily mean temperature exceeds the calendar day 90th percentile. This relative approach to define HWs captures events at all latitudes including those events in colder climates and during local winter months. This ensures that we capture events which can have significant impacts due to system resilience and adaptation even if absolute temperatures might not be high.”

An issue which has also not been discussed in this context is the role of time scales. The authors consider daily events of, e.g., high temperatures and low stream flow. I doubt that such an event of a single day length would really cause a severe impact. The authors should discuss the relevance of their selected hazards in more detail and really clarify their impact relevance.

Most multivariate CEs included in this study can have impacts on very short time scales, i.e. hours. Example for this are CEs potentially leading to floods, damage related to storms (e.g. wind, water and/or hail damage), and bushfires. For the reviewer’s example of high temperature and low streamflow the duration of one day can already lead to an economic impact in the sense that riverine transport is likely to be delayed which leads to a delay in shipping and thus extra costs for storage and refrigeration due to the higher temperatures.

We address our choice to focus on single day events in the discussion of the manuscript. For this we added the following statement:

“Our decision to focus on CEs using a daily time step provides us with a rich sample to calculate the co-occurrence of pairs of hazards. We recognise that events with the duration of 1 day will not always cause significant socio-economic impacts for all hazard combinations considered in this study. Adding a temporal dimension in the analysis is an area worthy of future work, but we note that it would increase the complexity quite substantially and reduce the data available for analysis.”

For other hazards the relevance of the compoundness is not really clear to me. For instance, I am not sure what is so relevant about co-occurring heatwaves and large hail events. The impacts are, to my understanding, quite different, such that no compound impact may be caused such as in the case of compound flooding or heat and drought. If there is such an impact, it should be explained better.

We acknowledge that the possible consequences of some hazard combinations might not be immediately obvious. However, we indicated possible impacts in Supplementary Table S1 and discuss some potential impacts in the context of the results. The specific example given by the

reviewer (heatwave and hail) may for example exacerbate crop damage. In our revision, we added the following sentences to the main text to guide the reader:

“[...] potential to cause socio-economic impacts. **The possible impacts of some hazard pairs are more obvious than others. Supplementary Table S1 presents a list of possible socio-economic impacts including less obvious potentially impact-bearing hazard combinations, e.g. the joint occurrence of low streamflow and heatwave which can lead to increased transport costs due to shipping delays and the requirement of additional refrigeration and storage^{21,22}. Other combinations might cause a joint impact in the sense of monetary loss due to crop failure caused by heatwaves and/or drought conditions paired with hail damage of crop and/or property in the same region.**”

- 21 Wilkes, W. *Rhine River Shipping Faces Another Historic Shutdown as Drought Hits Water Levels*, <<https://www.insurancejournal.com/news/international/2019/07/24/533767.htm>> (2019).
- 22 Ellyatt, H. *A major river in Europe hit by drought could create economic havoc*, <<https://www.cnbc.com/2019/07/31/low-water-levels-in-the-river-rhine-could-create-havoc-for-germanys-economy.html>> (2019).

An issue which needs to be made transparent is the fact that the study is not purely observational. E.g., winds and hail are taken from reanalysis data. As long as a sufficient amount of observational data has been assimilated, this is not necessarily a problem. But for hail and similar variables, the output is a pure model product even based on parameterisations (such as cloud microphysics). Thus these data cannot be used as an independent observational source to evaluate models. This shortcoming needs to be made explicit and transparent in a prominent place (e.g., the introduction). Otherwise it just corroborates the impression that the authors oversell the relevance of their study.

We agree – we did not intend to hint that reanalyses are independent observations but they are of course useful. We addressed by adjusting the sentence introducing the study in the introduction:

“We combine 12 different hazards **from observations complemented with reanalyses** to form 27 hazard pairs with the potential to cause **ecological and** socio-economic impacts.”

and in the discussion of the manuscript by adding the following:

“We present a first global climatology for a wide range of compound events with the potential to cause severe socio-economic impacts **based on observations and reanalysis products** spanning over more than three decades (1980 – 2014). **We used re-analysis data to supplement observations; not all hazards could be derived from observational data due to the lack of available data products (e.g. wind speed and wind-related hazards like storm surge and wave height). In many parts of the world, the reanalysis products are constrained by high quality observations and reproduce these well. However, reanalyses are poorly constrained in data sparse regions and inevitable variations exist in how reliable our estimates of the joint probabilities of hazard pairs would be. We would therefore encourage a repeat of our analysis with alternative observations and newer reanalyses when they become available. However, while inevitably limited [...]**”

Further, in the Supplementary Tables we added an asterisk to each variable derived from reanalysis data.

The writing of the manuscript has been somewhat sloppy and needs substantial improvement, see also the specific comments. Regarding structure, the authors need to add a link between the general introduction and the results part. Here the main concepts (e.g., how a hotspot is defined, what type of return period has been chosen) need to be at least briefly presented, with explicit links to the relevant methods sections. Otherwise the reader is lost.

To resolve this, we added the following sentences at the beginning of the Results section:

“In the following, CE hotspots are defined as geographical regions with short return periods in the joint occurrence of a specific hazard pair. In the multivariate context of CEs chosen in this paper, return periods (hereafter joint return periods; RP) are based on the probability that both hazards in a given pair exceed their threshold simultaneously (see Methods). The geographical joint occurrence of key hazard pairs (see list in Supplementary Table S1) is shown in [...]”

Further we adjusted the Methods to include a definition of return periods earlier in the section. The beginning of the third paragraph now starts with:

“Results are presented as return periods, i.e. the inverse of occurrence probability which in this study is determined as follows. If both hazards [...]”

The usage of the term "risk" is not very clear. The authors state (l 249) "Risk is defined as the probability of hazards and represents a convolution of the respective distributions of probability and consequences.". Isn't this self contradictory, as the hazard is only defined by a probability? A more useful definition, also in line with the IPCC framework (and I think this is what the authors mean) would be to define risk as stated in the last sentence, but then it is not the probability of a hazard, but relates to the impact.

We agree with the reviewer that this sentence is confusing. As it is not necessary for the understanding of our study or the applied methods we removed this sentence entirely. Further we replaced the word “risk” throughout the manuscript when it was not used in the context of the IPCC framework. For example, we changed “increased risk of flooding” to “increased **probability of flooding**” and “increased wildfire risk” to “increased **fire danger**”.

Overall the study is, as stated above, very comprehensive and definitely useful for the broad readership of Nature Communications. As discussed, it has important limitations because impacts have not been directly considered. This is not the author's responsibility, almost all large-scale studies on compound events suffer from this shortcoming, simply because only few suitable large-scale impact observations exist. But this limitation should be made transparent. In many places, I have the feeling the authors oversell the relevance of their study. In fact, I am wondering whether the study merits publication in Nature Communications. The authors have made a big effort to collect data and to calculate compound hazards. But the study lacks originality beyond rather the straight forward analyses.

We thank the reviewer for their encouraging judgement. We are confident that the revised manuscript has addressed the shortcomings in a transparent manner and will be useful for the broad readership of Nature Communications. We obviously do not agree with the Reviewer's questioning of the merits of our paper for Nature Communications. As the reviewer noted, our

“study is comprehensive and thus very useful for a broader community”. Nature Communications is an ideal forum to communicate with the broader community – and publishing this paper in (say) Journal of Climate might not achieve this outcome. Our paper is also the first attempt at a global determination of compound events and as noted by Reviewer 2 is “an important first step”. We hope it will provide a foundation for a great deal of work underway across the community.

Specific Issues

37 does "generate" really refer to "interaction", as the grammar suggests, or rather to the "drivers" (then an "s" is missing)

Yes, the subject in the sentence is interaction therefore the verb needs to be in third person singular.

46 I do not fully agree with this statement. If one had an impact time series, one could well do a univariate analysis as well. Only to understand the interplay, a multivariate analysis would be required, or in case no impact data was available (i.e., in a prediction context). The authors rather want to point out that the co-occurrence of two events is underestimated if the dependence is not modelled, but this is a different thing, and the authors should explain that precisely.

We partly agree. Without a significant interaction/correlation between drivers, univariate approaches could be used. We revised the sentence as:

“CEs are by definition events with multiple, potentially interacting, meteorological processes and consequently require different analysis methods compared with their univariate counterparts.”

However, we disagree that an impact time series combined with a univariate analysis would be enough. If multiple drivers are relevant, even in the fortunate case when data on impacts are available, we would not be able to characterize the event fully through a univariate analysis.

54 Also study 17 should be mentioned as a large-scale study of compound flooding - all the European coastlines have been investigated.

Done.

59 The authors slightly oversell the relevance of their study here - why does the lack of similar studies prevent the development of a better understanding of the mechanisms underlying compound events? I really don't buy that. Mechanisms causing, e.g., compound flooding or dry heat waves are in principle well understood. What is new is just the quantification.

We appreciate the reviewers concern that this statement might be unclear. We do however argue that the mechanisms causing different CEs are not necessarily well understood. While this might be the case for specific events of the past, a general overview is not readily available particularly beyond the regional scale. We adjusted the text as follows:

“No previous analysis has examined links between a range of hazards, or the geographic regions where different CEs are most likely to occur. **Instead, studies have focused on describing specific events, the impact of correlation on return periods² and, with the exception of a few isolated studies²⁰, regional scales.**”

69 Delete the "robust" - it is not needed here.

Done.

71 how does the dataset help to study the mechanisms behind CEs? I find this a bit far-stretched. The same holds even more for assessing the reliability of climate models to predict future changes in CE occurrence. This is important, but it requires process-based analyses.

We adjusted the manuscript as follows:

“This lack of a global geographical fingerprint of CEs limits the ability to design studies to better understand the mechanisms underlying CEs and to assess, plan for and mitigate the consequences of CEs.”

75 there is no unique definition of return periods for the multivariate case (see e.g., Serinaldi 2015, or the discussion in Bevacqua et al., 2019). In the methods section, the type is specified (AND), but here should be a reference, and somewhere the choice should be explained, as it affects the results.

A more detailed discussion of the concept of return period in the context of multivariate extremes was introduced at the beginning of the results section (see response to previous comment). The added text is:

“In the following, CE hotspots are defined as geographical regions with short return periods in the joint occurrence of a specific hazard pair. In the multivariate context of CEs chosen in this paper, return periods (hereafter joint return periods; RP) are based on the probability that both hazards in a given pair exceed their threshold simultaneously (see Methods).”

And in the methods section:

“[...] Information about the coverage of different hazard pairs are shown in Supplementary Fig. S9. We note this is one specific combination of bivariate probability, i.e. both hazard X and hazard Y: $P(X > x \wedge Y > y)$. However, multivariate risk assessment can incorporate more than multivariate CEs and depending on the impact one may consider a combination of different joint probability described by a variety of different possible hazard combinations⁴, e.g. both hazard X and hazard Y: $P(X > x \wedge Y > y)$ and hazard X or hazard Y: $P(X > x \vee Y > y)$.⁴⁶ For the purpose of this paper this is however not appropriate due to the choice to analyse exclusively multivariate CEs.

Global compound event hotspots are determined by focusing on the identification of hotspots for each CE type/hazard pair individually. [...]

244 this section is not well organised. E.g., return periods are used implicitly throughout the section, but they are only defined very far to the end. The discussion of land-based hazards and storm surges appears just out of nowhere - is this the best place to put it?

We reorganised this paragraph to improve the reading flow and moved the definition of return period to the beginning of this subsection (refer to response to previous comment).

“Results are presented as return periods, i.e. the inverse of occurrence probability, which in this study is determined as follows. If both hazards in a pair jointly exceed their respective threshold on the same day in the same grid cell the day is marked as a CE. **Land-based hazards, storm surge and wave height do not share the same grid cells, with the latter occurring only over the sea. To identify CEs formed by a combination of ocean-based and a land-based hazard, coastal grid cells are defined as land grid cells with at least one neighbouring ocean grid cell. As an example for the combination of two land-based hazards, a day is considered a compound event of the type wind and precipitation** if the total precipitation and the daily maximum wind speed on that day both lie within the top 1% of events during the study period. In terms of occurrence probability this means that if the two hazard pairs are independent the probability of their joint occurrence should be $0.01 \times 0.01 = 0.0001$ which is equivalent to roughly one occurrence during the study period (12,874 days). **To identify CEs containing surge and/or wave height** each coastal cell is linked to exceedances in surge in the four surrounding cells (both ocean and land cells with land cells having no values).”

258 are the percentile-based indices really relevant? I have been struggling myself here, but maybe the authors could help the reader a bit by giving some advise on the limitations of this approach. Consider a "heatwave" with temperatures above 20C in a cool climate - such a heatwave will cause very different impacts - if at all! - than a heatwave of 40C in a hot climate. This further highlights the need to discuss the "potentiality" character of your study.

We would like to note that the identification of heatwaves used a percentile threshold and not an absolute threshold as implied by the reviewer and uses the well-established and widely used Excess Heat Factor. The inclusion of the percentile of temperature is relevant due to the fact that most systems are adapted to the prevailing climatological mean conditions. Therefore, every deviation from the mean puts stress on the system, particularly if this deviation is large and sustained. Therefore, in case of the heatwave example the reviewer refers to, the system in the cool climate may be equally stressed as the one in the hot climate. A negative impact is therefore likely to occur even if the absolute temperatures are low (see specific examples in our response above).

360 This section is very sloppy. For a significance test, you need to clearly define a null hypothesis. What is the null hypothesis here?

We added the following sentence to the beginning of the section describing the significance analysis.

“A significance test is applied for each hazard pair to assess whether the joint exceedance of a hazard pair is significant at a specific grid-cell. We test the null hypothesis that the joint return period of the hazard pair found in each grid cell can be reproduced by chance and does not require any physical correlation between the two hazards. For this, [...]”

491 Reference 17 should be updated (see below in the list of references)

Done.

Supplementary Information:

The preparation of the supplementary information has been very sloppy. No author list is provided, and the page breaks are sometimes annoying. E.g., Table 1 ends just after the break, so that it is not quite easy to find the caption.

We adjusted the type setting of the Supplementary Information and added an author list.

References

E. Bevacqua, D. Maraun, I. Hobaek Haff, M. Widmann and M. Vrac: Multivariate statistical modelling of compound events via pair-copula constructions: analysis of floods in Ravenna (Italy), *Hydrol. Earth Syst. Sci.*, 21, 2701–2723, 2017.

E. Bevacqua, D. Maraun, M.I. Voudoukas, E. Voukouvalas, M. Vrac, L. Mentaschi, and M. Widmann: Higher potential compound flood risk in Northern Europe under anthropogenic climate change, *Science Advances* 5, eaaw5531, 2019.

F. Serinaldi: Dismissing return periods! *Stoch. Environ. Res. Risk Assess.* 29, 1179–1189, 2015.

Reviewer #2

REVIEWER COMMENTS

Review of NCOMMS-20-09682-T, Global hotspots for the occurrence of compound events, by Ridder et al.

Ridder and colleagues present a global climatology of compound extremes by analyzing a set of combined climate variables and hazards co-occurring simultaneously. The analysis is an important first step and I'm generally supportive of the manuscript. The challenge is the amount of information presented in the figures and the supplement, and the fundamental lack of space in a letter format to give the results the due diligence they deserve given the work these authors have done. I leave it to the editor to decide whether this is an appropriate format, but I note that there are 16 supplementary figures, which strikes me as a lot. Regardless of that, I have four principal comments along methods and CE philosophy that the authors might consider as they revise the manuscript.

Methodology:

1. Data uncertainty: The authors have done a nice job corralling a lot of data and being thoughtful about the compound pairs they analyze. One practice that is emerging however, particularly when estimating at the tails of a joint distributions in observations and reanalyses, is the use of multiple datasets to provide an observational ensemble (e.g., Horton et al., Nature 2015; Coffel et al. Earth's Future 2019). Given the rigor of some of the analyses (e.g., field significance, which needs to be employed much more in climate studies), I was surprised the authors were content with using a single data source for each variable, especially given the fine temporal resolution of their analysis and their emphasis on building the first climatology of the co-occurrence of extremes. This sampling uncertainty can be quite large and while the bootstrapping addresses some portion of this uncertainty, it rests on the assumption that the sample is representative of the population, which it may not be. I would encourage the authors to include more observational and reanalysis datasets in their analysis.

We thank the reviewer for raising this issue. We carefully chose the datasets used in this study taking into account their quality and ensuring global coverage. Further, we aimed for consistency in the source of data to preserve the connection between different hazards and ensuring the collocation of observational points as much as possible. While the choice of a single data source for each hazard makes our results somewhat sensitive to the chosen datasets we believe our results in their current form present a sufficient foundation to inform policy makers and stakeholders concerned with mitigating the impacts of CEs. We added the following statement to the Discussion to highlight the issue and advise a follow up study comparing our results with those derived using different observations and newer reanalysis data.

“However, reanalyses are poorly constrained in data sparse regions and inevitable variations exist in how reliable our estimates of the joint probabilities of hazard pairs would be. We would therefore encourage a repeat of our analysis with alternative observations and newer reanalyses. However, while inevitably limited by data coverage and uncertainties in both observational and reanalysis data, the regionally dependent hotspots identified here, and the varying regional importance of different hazard pairs provide a foundation to inform policy and stakeholders concerned with mitigating the impacts from CEs.”

We also added the following justification for the choice of dataset throughout the Methods section:

“We consider a total of 12 different land-based hydro-climate variables and indices, namely daily high and low precipitation sums, Standardised Precipitation Index (SPI) as a metric for meteorological drought, high and low streamflow, daily maximum temperatures, the Excess Heat Factor (EHF) as a measure for heatwaves, the McArthur Forest Fire Danger Index (FFDI) to quantify fire weather conditions, the probability of hail, maximum wind speeds, maximum storm surge, and maximum wave heights. **The sources for the different hazards were chosen to ensure use of high-quality data, consistency of observation locations and to ensure global coverage of the data as far as possible. Details on each dataset are listed below. [...]** Daily maximum temperatures were taken from the HadGHCND observational dataset.³⁶ **HadGHCND is the largest available repository of global daily in-situ observations for temperature. It has been specifically designed to analyse extremes that has been relied upon in past studies^{37,38} which makes this dataset particularly suitable for the purpose of this study. [...]** Daily precipitation sums were derived from the 1°x1° observational dataset REGEN.³⁹ **This dataset represents a combination of two of the largest in-situ observational repositories, namely GPCP and HadGHCND. This reduces biases and uncertainties associated with single source datasets while incorporating information from the same observational system as the temperature dataset chosen here. As such REGEN is highly suitable for the analysis of compound events in the context of this study. ”**

2. Percentile estimation: For those climate variables where percentile estimates are used, how are those estimated? There is no information provided on the distribution fitting methods or parametric assumptions, which is another major source of uncertainty not included.

We added this information to the method section at the end of the second paragraph in the subsection “*Definition of Compound Events and hotspots*”.

“[...] This is either a fixed value for the hazard indices EHF, SPI, hail probability, and FFDI or a percentile threshold for climate variables. Percentiles are determined for each grid cell individually to account for regional differences (see Supplementary Table S2 for thresholds). **Percentile values were derived using the Python NumPy function “percentile”, which estimates the value of a specified percentile using provided data points (e.g., a value was associated with the 90th percentile if approximately 90% of the data points were equal to or less than that value). Linear interpolation was used to determine percentile values when they fell between two data points.”**

CE philosophy:

3. Definitional: I recognize one of the co-authors conceived of the definition of compound extremes as either variables and/or hazards. But I wondered about the focus on only those that were simultaneous. Certainly a compound pair would be impactful if they were back to back?

CEs come in different types as described by Zscheischler et al. (2020). For this study we chose to focus on co-occurring hazards in space and time, which have received by far the most attention in recent literature. We indicated this by stating: “**One specific class of CEs, multivariate events⁴, occurs when two or more drivers and/or hazards impact a region simultaneously.**” in the introduction which was also included in the original version of the manuscript. To acknowledge

that other types of CEs exist and explain our choice to focus on multivariate CEs we added the following statement to the discussion section:

“The class of multivariate compound events analysed in this study is only one of the possible types of CEs, all of which have potential to cause socio-economic impacts. We selected multivariate compound events because this type of CEs has received the most attention in recent years⁴ and climatology of these events is therefore timely. Our decision to focus on CEs with a duration of day provides us with a rich sample to calculate the co-occurrence of pairs of hazards.”

4. Impact: My previous comment, I think, links to one of the main things that is missing from the analysis, which is the actual impact. The analysis assumes that these percentiles or occurrences are themselves impactful to people and their socio-economic systems. The actual impacts of these things are divorced from the analysis presented, which is really problematic from an interpretation standpoint. A hotspot may emerge as a hotspot by this analysis, but there may not be any actual consequences of being a hotspot—meaning the compound pair does not actually have socio-economic impacts. If the point is to aid risk assessment, then a glaring omission is the assumption that a top-down approach like this is actually meaningful in informing risk, without some assessment of how often such CEs actually select for on-the-ground impacts. I would encourage the authors to think about how their extreme occurrence relates to outcomes they seek to inform the risk management of. Consider the recent work of Kornhuber et al. Nat Clim Chg 2020, where they linked quasi-stationary wave occurrence to actual agricultural impacts. The scientific community, if it is actually interested in informing risk assessment, must move beyond assuming that a 90th percentile heat event is problematic and instead show that it is problematic.

We thank the reviewer for highlighting this issue and agree that a clear difference between CEs and socio-economical impact needs to be made. We therefore adjusted our wording throughout the manuscript to highlight that CEs only have the potential to cause impacts and not all CEs trigger impacts. To clarify this distinction, we adjusted the manuscript where necessary. We adjusted the first sentence in the Summary to read:

“Compound events (CEs) are weather and climate events that result from multiple hazards or drivers with the potential to cause severe socio-economic impacts.”

Further we changed the first paragraph of the Introduction to:

“Extreme weather and climate events often result from a combination of multiple hazards or drivers. These events are commonly referred to as Compound Events (CEs).^{1, 2, 3, 4} The interaction of multiple hazards and/or drivers that generates CEs often leads to more severe ecologically and socio-economically damaging events compared to single-hazard events.⁵ Multivariate risk assessment sometimes requires multivariate extremes to have an impact to be considered a CE. Here we use the risk framework of the Intergovernmental Panel for Climate Change (IPCC) in which extreme weather and climate events are considered a hazard based solely on the variable’s value compared to its distribution noting that rare meteorological phenomenon may not necessarily cause damage.⁴”

“[...] high fuel load and strong winds. Adopting the IPCC’s definition of extreme climate events means that not all occurrences of a CE lead to an impact as these are dependent on a

combination of hazard occurrence as well as vulnerability and exposure of the affected region/system. Our analysis therefore focusses on the occurrence of CEs defined as the joint probability of two hazards and we do not explore whether the CEs necessarily lead to impacts. For multivariate risk assessments that require CEs to have impacts, our results can be interpreted as the climatology of the precursors to CEs.”

We also added the following statement to the Discussion and Conclusions:

“[...] the varying regional importance of different hazard pairs provide a foundation to inform policy and stakeholders to mitigate impacts from **multivariate CEs**. **A direct translation of the CE hotspots identified in this study into impact maps is however not possible due to the differentiation between risk and impact and the here applied definition of compound events.**² In the IPCC framework, which we adopt for this study, risk is defined as “the potential for adverse consequences of a climate-related hazard” and results from the combination of hazard, vulnerability and exposure.^{2, 4} Impacts are the consequence of risk without being interchangeable with risk. As such, while CEs as weather and climate hazards contribute to risk, they can occur without resulting in an impact. Their occurrence probability, however, is an indicator for the possibility of adverse impacts.

We hope these modifications to our manuscript will please Reviewer 2 – we recognise the effort they put into their review and we have tried hard to ensure their advice is reflected in the revised manuscript.

Reviewers' Comments:

Reviewer #1:

Remarks to the Author:

Review of "Global hotspots for the occurrence of compound events"

The reviewers have addressed many of the issues I raised, but I still have three major issues.

First, as stated in my first review, I am not sure the study is suitable for a high impact journal such as NCOMMs. The analysis is not original in its idea, straight forward to conduct based on standard datasets (which, as noted by the authors, in several cases may not be appropriate), provides useful but not exciting results. I would see "Weather and Climate Extremes" or maybe "Environmental Research Letters" as more appropriate.

The writing is still very sloppy, the text full of track changes, clumsy sentences, terms which have not been introduced, references to definitions which are not explained. Please find a non-comprehensive list below. The rebuttal letter is of a similar "quality". I am somewhat concerned that not only the writing but also the analysis may suffer from a lack of care and diligence.

I am still not quite satisfied with the framing of the whole study when it comes to the connection between the results and actual compound events. Zscheischler et al. (2018, cited by the authors) define "compound compound weather/climate events as the combination of multiple drivers and/or hazards that contributes to societal or environmental risk". The link between risk and impact, which the authors discuss, is completely irrelevant here as the difference between the two is merely the probabilistic character of risk whereas an impact means a risk has materialised. The reason why I suggested to include the term "potential" here is that it is not even clear whether the events analysed in the study contribute to risk. Without a study of exposure and vulnerability, it is impossible to answer whether these events contribute to any risk in a particular region.

Further Issues:

I 36/37: What is a driver?

I41 - 44: clumsy

I52: what is the IPCC's definition of extreme climate events?

I55: what is to be deleted here?

I65: reference 13 is not about univariate events, but about compound events. It should be listed in line 67.

I71: reference 19 is not a model study. It should as well be cited in line 67.

I75: a climatology is not the same as a fingerprint. Please be precise here.

226-229: "A direct translation of the CE hotspots identified in this study into impact maps is however not possible due to the differentiation between risk and impact and the definition of compound events applied here" - I

disagree with this statement. The reason is not about differences between risk and impact, but because of the very nature of impacts and that one would either need observations of impacts, or a function that translates hazards into risks. In the following sentences, I am afraid, the authors are also grossly misinterpreting the IPCC risk framework. Talking of impact maps is not useful because, as long as a hazard has not manifested itself, a risk will not manifest as an impact. Of course a hazardous event may not always cause an impact - but the key point is that the event may not occur.

244: "and climatology of these events is therefore timely" - this is not a complete phrase.

251-255: these two sentences are not quite precise. What does "For some" refer to? Hotspots? CEs? Researchers? And do you mean precursors, or preconditions? The former has a strong temporal connotation.

References: several references are not complete.

Reviewer #2:

Remarks to the Author:

Review of NCOMMS-20-09682-A, Global hotspots for the occurrence of compound events, by Ridder et al.

Ridder and colleagues present a substantive revision of their original submission. I think the manuscript is a nice contribution and with some very minor text revisions, should be published. Below are the original points I made and my response to their revision.

Methodology:

1. Data uncertainty: The authors have skirted this question a bit, which is entirely fine given the scope of the work they've performed. But I was hoping to get a sense of how sensitive their results are to the data choices they made. Even for a single CE, one could easily show how sensitive the results are to data combination. But given that there were 12 (now 10) supplementary figures, it seems appropriate to leave that to follow on work. But given this, I would respectfully disagree with the authors' claim at line 223 in the revised manuscript: "while inevitably limited by data coverage and uncertainties in both observational and reanalysis data, the regionally dependent hotspots identified here, and the varying regional importance of different hazard pairs provide a basis to inform policy and stakeholders concerned with mitigating the impacts from multivariate CEs." This is because the uncertainty from data choice is not at all assessed, so there is really no basis for the claim of a "sufficient" foundation to inform, without that potentially being misinformation. So I would suggest deleting that sentence and the other claims in the paper that say this work represents a sufficient basis to inform policy. See also my point 4 about impacts, below. This work is important and worthy of publication even without that claim being true.

2. Percentile estimation: Generally addressed by the authors.

CE philosophy:

3. Definitional: Addressed by the authors.

4. Impact: R1 and I both agree that the analysis is divorced from actual impacts. That in of itself is not problematic, it just limits the claims to being policy-relevant. The authors have appropriately tried to adjust their language accordingly. But I still think there is room to make some minor text adjustments in the same vein as the data uncertainty point I make above: There is no real basis to

inform stakeholders or policy because the risks of CE occurrence alone an impact does not make. One can have large changes in CE occurrence but no actual impacts associated with them. Like point 3, I would ask the authors to briefly revisit the textual claims to the risks of occurrence themselves being sufficient to inform policy.

Thanks for the opportunity to review and I hope all of the authors and their families are safe and healthy during this trying time.

We thank both reviewers for their advice, detailed comments and suggestions. We have addressed each of their comments carefully and adjusted the manuscript accordingly. Below you will find our response to each reviewer comment. The reviewer's remarks are in black, our response is highlighted in blue and extracts from the manuscript are in red, with new text that has been added marked in bold.

Kind regards,

Nina Ridder (on behalf of all authors)

REVIEWER COMMENTS

Reviewer #1 (Remarks to the Author):

The reviewers have addressed many of the issues I raised, but I still have three major issues.

Thank-you for your positive comments on how we have addressed many of the issues you raised. We note you have three remaining issues and we address these individually below.

First, as stated in my first review, I am not sure the study is suitable for a high impact journal such as NCOMMs. The analysis is not original in its idea, straight forward to conduct based on standard datasets (which, as noted by the authors, in several cases may not be appropriate), provides useful but not exciting results. I would see "Weather and Climate Extremes" or maybe "Environmental Research Letters" as more appropriate.

We acknowledge the reviewer is not convinced that our paper is suitable for Nature Communications. The submission to Nature Communications was recommended by Nature Climate Change where the editor noted the likely high impact of our paper, its novelty and the need to communicate it to a broad readership. Nature Climate Change quite reasonably noted our work lacks the climate change lens. We took the advice of that Editor, and we note your concerns are not shared by Reviewer 2.

It is, of course, a judgment call and we would obviously respect the decision by the Nature Communications editor. We would note that research on compound events is relatively new, high profile, high impact and now forms the basis of major multi-disciplinary research groups in the US and Europe. Our paper provides the foundation for a lot of this research, and provides the key to unlock new research directions simply because a climatology of the form we provide has never been attempted before. Our view is this is therefore a topic of high relevance for Nature Communications, and that is a view clearly shared by Reviewer 2.

We think it a little unfair that the reviewer hints that our paper might be worrisome because of the difficulties in using some standard data sets. We think it is important to raise these issues clearly in a paper. Criticising honest appraisal acts to discourage open discussion of caveats. We also think it is a little unfair that the reviewer hints at a lack of originality in the "idea" we provide here. It is new, novel, and it has never been done before. It may be that elements of the methods are

standard, and the individual data are standard, but putting them together in the way we do is novel and we think exciting and we hope readers of Nature Communications would think so too.

The writing is still very sloppy, the text full of track changes, clumsy sentences, terms which have not been introduced, references to definitions which are not explained. Please find a non-comprehensive list below. The rebuttal letter is of a similar "quality". I am somewhat concerned that not only the writing but also the analysis may suffer from a lack of care and diligence.

This criticism is unfair and the hint at the end is not a reasonable conclusion to reach. We fully accept there were three words left in the paper that had been struck-through and not deleted when all track-changes were accepted for the "clean" version. This was sloppy, but hardly lends support to criticise the overall paper. Several of the "non-defined terms" are commonly used and we did not consider definitions needed given length considerations (although we have compromised and made the changes requested). There may also be occasional clumsy sentences but there are not many and it is helpful when a reviewer highlights this and we welcome the opportunity to revise them.

In short, we accept the criticism of three words and one coloured letter accidentally left in the final manuscript but we reject the implications the reviewer draws from this. We have, however, undertaken a careful review of all the language by an experienced editor to address any possible remaining concerns.

I am still not quite satisfied with the framing of the whole study when it comes to the connection between the results and actual compound events. Zscheischler et al. (2018, cited by the authors) define "compound weather/climate events as the combination of multiple drivers and/or hazards that contributes to societal or environmental risk". The link between risk and impact, which the authors discuss, is completely irrelevant here as the difference between the two is merely the probabilistic character of risk whereas an impact means a risk has materialised. The reason why I suggested to include the term "potential" here is that it is not even clear whether the events analysed in the study contribute to risk. Without a study of exposure and vulnerability, it is impossible to answer whether these events contribute to any risk in a particular region.

We are unclear as to the point the reviewer makes here as we took their earlier advice and added the word potential as asked to do. We do acknowledge a few specific sentences that warranted editing and we discuss those below.

Further Issues:

l 36/37: What is a driver?

The word driver is in common use and we did not think it needed to be defined. However, we are happy to define it and have added the following explanation to line 36/37 to make clear what we consider as a driver:

"Extreme weather and climate events often result from a combination of multiple hazards or drivers (a driver is a direct cause of climate-related hazards, see table 1 in 1). These events are often referred to as Compound Events (CEs)^{1, 2, 3, 4.}"

1. Zscheischler J, *et al.* Future climate risk from compound events. *Nature Climate Change* **8**, 469-477 (2018).

I41 - 44: clumsy

We apologise if the reviewer finds this part “clumsy”. The original text was:

“Here we use the risk framework of the Intergovernmental Panel for Climate Change (IPCC) in which extreme weather and climate events are considered a hazard solely based on the variable’s value compared to its distribution noting that rare meteorological phenomenon may not necessarily have consequences.”

We think the meaning is clear in the original. However, we repeat this statement in a similar way a few lines below. We therefore removed this sentence from the manuscript as we believe the sentences in lines 48 – 53 outline the adopted framework sufficiently.

I52: what is the IPCC’s definition of extreme climate events?

We added the following explanation to the text to clarify this:

“**Here we use the risk framework of the Intergovernmental Panel for Climate Change (IPCC) and define climate extremes** as the occurrence of a value of a weather/climate variable within either tail of the variable’s observed distribution⁴. **This means** that not all occurrences of a CE **necessarily** lead to an impact as these are dependent on a combination of hazard occurrence as well as vulnerability and exposure of the affected region/system.”

4. SREX I. Managing the risks of extreme events and disasters to advance climate change adaptation. *A Special Report of Working Groups I and II of the Intergovernmental Panel on Climate Change*, edited by: Field, CB, Barros, V, Stocker, TF, Qin, D, Dokken, DJ, Ebi, KL, Mastrandrea, MD, Mach, KJ, Plattner, G-K, Allen, SK, Tignor, M, and Midgley, PM, Cambridge University Press, Cambridge, UK, and New York, NY, USA, (2012).

I55: what is to be deleted here?

We apologise that we overlooked this in our first review of the manuscript. We corrected this and the sentence now reads:

“Our analysis focusses on the occurrence of CEs defined as the joint probability of two hazards and we do not explore whether the CEs necessarily lead to impacts.”

I65: reference 13 is not about univariate events, but about compound events. It should be listed in line 67.

We have made this modification

l71: reference 19 is not a model study. It should as well be cited in line 67.

Agreed, and we have made this modification.

l75: a climatology is not the same as a fingerprint. Please be precise here.

We have changed the text to:

“climatological fingerprint”

226-229: *"A direct translation of the CE hotspots identified in this study into impact maps is however not possible due to the differentiation between risk and impact and the definition of compound events applied here"* - I disagree with this statement. The reason is not about differences between risk and impact, but because of the very nature of impacts and that one would either need observations of impacts, or a function that translates hazards into risks. In the following sentences, I am afraid, the authors are also grossly misinterpreting the IPCC risk framework. Talking of impact maps is not useful because, as long as a hazard has not manifested itself, a risk will not manifest as an impact. Of course a hazardous event may not always cause an impact - but the key point is that the event may not occur.

We do not think our text was “grossly misinterpreting the IPCC risk framework” but equally we would agree with the sentiments of the reviewer that we need to make absolutely certain that we do not mislead our audience.

To address this we revised this sentence and replaced the term “impact maps” with “risk maps”. We further edited our comments about the connection between hazards, risk and impacts:

“While a translation of the CE hotspots identified in this study into risk maps would be useful for policy decision making, it falls outside of the scope of this study as this requires additional information about regional vulnerability and exposure in accordance with the IPCC framework^{1, 4}. The identified CEs, however, are an indicator for the possibility of adverse impacts. Our results are therefore an important first step and represent a foundation for the analysis of actual impacts caused by the different CEs assessed in this study.”

244: "and climatology of these events is therefore timely" - this is not a complete phrase. We added the indirect article “a” to this sentence to correct this. The sentence now reads:

“We selected multivariate CEs because this type of CEs has received the most attention in recent years³ and a climatology of these events is therefore timely.”

251-255: these two sentences are not quite precise. What does "For some" refer to? Hotspots? CEs? Researchers? And do you mean precursors, or preconditions? The former has a strong temporal connotation.

We changed the sentences to:

“We are confident that the **analysis of multivariate CEs in this study, and the hotspots identified, **are a necessary step forward for research into CEs. Our results provide (1) guidance on the occurrence probability of CEs which can be considered as precursors of actual compound event impacts, (2) information on the regional importance of CEs and (3) a climatology to evaluate how well global climate models simulate CEs.”****

References: several references are not complete.

We revised the references and completed where additional data was required and available.

Reviewer #2 (Remarks to the Author):

Review of NCOMMS-20-09682-A, Global hotspots for the occurrence of compound events, by Ridder et al.

Ridder and colleagues present a substantive revision of their original submission. I think the manuscript is a nice contribution and with some very minor text revisions, should be published. Below are the original points I made and my response to their revision.

Thank-you for your positive comments. We did undertake a substantive revision and we appreciate your recognition of this.

Methodology:

1. Data uncertainty: The authors have skirted this question a bit, which is entirely fine given the scope of the work they've performed. But I was hoping to get a sense of how sensitive their results are to the data choices they made. Even for a single CE, one could easily show how sensitive the results are to data combination. But given that there were 12 (now 10) supplementary figures, it seems appropriate to leave that to follow on work. But given this, I would respectfully disagree with the authors' claim at line 223 in the revised manuscript: *“while inevitably limited by data coverage and uncertainties in both observational and reanalysis data, the regionally dependent hotspots identified here, and the varying regional importance of different hazard pairs provide a basis to inform policy and stakeholders concerned with mitigating the impacts from multivariate CEs.”* This is because the uncertainty from data choice is not at all assessed, so there is really no basis for the claim of a “sufficient” foundation to inform, without that potentially being misinformation. So I would suggest deleting that sentence and the other claims in the paper that say this work represents a sufficient basis to inform policy. See also my point 4 about impacts, below. This work is important and worthy of publication even without that claim being true.

We agree with the reviewer that the information our results provide is not directly applicable to advise policymakers. To address the reviewer's concern regarding uncertainties and the sensitivity of our results to the choice of datasets, we have performed an assessment testing the

results of our global study for its validity for Australia. For this we used the local high-resolution dataset from the Australian Water Availability Project, as well as ERA5 reanalysis data. We managed to show that the main features for the most important multivariate CEs in this region were reproduced in the local datasets. We included a sentence pointing this out in the manuscript and added the relevant figure into the Supplementary Information.

“While our global results are limited by data coverage and uncertainties in both observational and reanalysis data, high-resolution regional datasets do exist that could be used to replace the global products. We therefore examined whether replacing the global datasets with alternative high-resolution data reproduce the main features of the global hotspots map for the most important CEs over Australia (Supp. Fig. S8). Our results show a high degree of similarity in the overall patterns of CEs using alternative data suggesting the regionally dependent hotspots identified in the global analysis, and the varying regional importance of different hazard pairs are robust. We therefore suggest that the global-scale analysis provide a useful first step in informing risk analysts and stakeholders concerned with identifying risk from multivariate CEs. While a translation of the CE hotspots identified in this study into risk maps would be useful for policy decision making, it falls outside of the scope of this study as this requires additional information about regional vulnerability and exposure in accordance with the IPCC framework^{1, 4}. The identified CEs, however, are an indicator for the possibility of adverse impacts.”

Supplementary Figure S8 Comparison between CE hotspots in Australia derived from global datasets (left; see Supplementary Table S2 for details) and regional high-resolution datasets (right). Shown are the most important hazard combinations for this region. Regional high-resolution data for precipitation, drought (SPI) and heatwaves (EHF) were taken from the Australian Water Availability Project (AWAP)¹. High-resolution 10m winds were derived from

ECMWF's ERA5's zonal and meridional 10m wind components. High-resolution FFDI data were taken from the Copernicus Emergency Management Service for the European Forest Fire Information System (EFFIS)². Streamflow data were derived from a high-resolution gridded version of the global dataset^{3, 4}.

1. Raupach M, Briggs P, Haverd V, King E, Paget M, Trudinger C. Australian water availability project. *Canberra: CSIRO Marine and Atmospheric Research* (2012).
2. Vitolo C, *et al.* ERA5-based global meteorological wildfire danger maps. *Scientific data* **7**, 1-11 (2020).
3. Do HX, Gudmundsson L, Leonard M, Westra S. The Global Streamflow Indices and Metadata Archive (GSIM)-Part 1: The production of a daily streamflow archive and metadata. *Earth System Science Data* **10**, 765-785 (2018).
4. Gudmundsson L, Do HX, Leonard M, Westra S. The Global Streamflow Indices and Metadata Archive (GSIM)-Part 2: Quality control, time-series indices and homogeneity assessment. *Earth System Science Data* **10**, 787-804 (2018).

2. Percentile estimation: Generally addressed by the authors.

Thank you.

CE philosophy:

3. Definitional: Addressed by the authors.

Thank you.

4. Impact: R1 and I both agree that the analysis is divorced from actual impacts. That in of itself is not problematic, it just limits the claims to being policy-relevant. The authors have appropriately tried to adjust their language accordingly. But I still think there is room to make some minor text adjustments in the same vein as the data uncertainty point I make above: There is no real basis to inform stakeholders or policy because the risks of CE occurrence alone an impact does not make. One can have large changes in CE occurrence but no actual impacts associated with them. Like point 3, I would ask the authors to briefly revisit the textual claims to the risks of occurrence themselves being sufficient to inform policy.

We thank the reviewer for giving us the opportunity to revise this. As mentioned in our response to point 1 we agree with the reviewer and we had tried to revise the manuscript with this in mind. However, we acknowledge there are a couple of places we could take one more step and we therefore revisited our manuscript and replaced our "policy" with "risk assessment" or removed references to policy completely.

"While our global results are limited by data coverage and uncertainties in both observational and reanalysis data, high-resolution regional datasets do exist that could be used to replace the global products. We therefore examined whether replacing the global datasets with alternative high-resolution data reproduce the main features of the

global hotspots map for the most important CEs over Australia (Supp. Fig. S8). Our results show a high degree of similarity in the overall patterns of CEs using alternative data suggesting the regionally dependent hotspots identified in the global analysis, and the varying regional importance of different hazard pairs is robust. We therefore suggest that the global-scale analysis provide a useful first step in informing risk analysts and stakeholders concerned with **identifying risk from multivariate CEs.”**

and

“Establishing those models with skill in simulating CEs, and those models that capture the right statistics with the right physical mechanisms, would provide planners and risk analysts clarity on which climate models are best suited to explore changing risks associated with CEs under a changing climate.”

Thanks for the opportunity to review and I hope all of the authors and their families are safe and healthy during this trying time.

We thank the reviewer for this and wish him/her all the best, health and strength to survive during this unprecedented time.